# Annual report text's positive tone and corporate green innovation: Evidence from China

**Yange Gao** [1]*, **Jian Feng** [2]

**1** Business School, Chengdu University, Chengdu, China, **2** Accounting School, Southwestern University of Finance and Economics, Chengdu, China

* gaoyange@cdu.edu.cn

## Abstract

From the perspective of annual report text information, we study the relationship between the annual report text's positive tone and corporate green innovation. Taking listed companies from 2010 to 2022 as a sample, we found that the positive tone of the annual report text significantly improves the company's green innovation while improving the quantity and quality of green innovation. The mechanism test shows that the main channels are easing corporate financing constraints and enhancing external attention. Regarding heterogeneity analysis, we found that the positive annual report text has a more significant effect on corporate green innovation in companies with high economic policy uncertainty and non-heavily polluting industries. Finally, we found that the positive tone of the annual report text can ultimately improve the company's long-term value through green innovation. Our study has enriched the theoretical research on the annual report text tone and provided empirical evidence for promoting enterprise green innovation.

**Data Availability Statement:** All relevant data are within the manuscript and its Supporting Information files.

**Funding:** This work was supported by the Tianfu Cultural Research and Cultural Innovation Project: Research on the Status quo and optimization of

## 1. Introduction

The report of the 19th National Congress of the Communist Party of China pointed out that it is necessary to build a market-oriented green technology innovation system, and the report of the 20th National Congress of the Party repeatedly mentioned "promoting green and low-carbon development." The 14th Five-Year Plan will also promote green and low-carbon development. In the practice process of green and low-carbon development, enterprises are the main body of natural resource consumption and pollution emission and an essential promoter of green governance and green innovation [1]. However, compared with general technological innovation, green innovation has "double externalities" [2, 3], coupled with the imperfect market system in China and the natural high risk, uncertainty, and long-term cyclical characteristics of green innovation, enterprises lack the internal drive for green innovation. Therefore, it is necessary to explore the factors that drive enterprise green innovation and guide the implementation of enterprise green innovation strategy. The existing literature on the influencing factors of green innovation is mostly based on government policies [4], industry competition

green development in Chengdu's construction as a World Cultural City [TYB202414]. The funders had no role in study design, data collection and analysis, decision to publish, or preparation of the manuscript.

**Competing interests:** The authors have declared that no competing interests exist.

pressure [5], stakeholder pressure [6, 7] and internal dimensions such as gender diversity of the board of directors [8] and overseas experience of senior executives [9].

In recent years, the use of modern analytical tools such as machine learning and text mining technology has refined the incremental information in the annual report in addition to financial information, which has well made up for the limitations of financial information that cannot reflect the overall situation of the enterprise and external development strategy, and provided a new perspective for the research on the pre-factors of corporate green innovation [10]. Previous studies have found that the positive tone of text language is valuable incremental information, which can convey to the outside world the company's future performance [11], stock market returns [12], social responsibility performance [13], and increase the investment intensity of external investors [14] and easing corporate financing constraints [15]. However, other studies have pointed out that the positive tone of text messages is not authentic and reliable. Corporate managers may deliberately use a positive tone to mislead investors to cover up the poor performance of enterprises in the environment and society [13]. Then, does the annual report text's positive tone affect the company's green innovation decision? What is the mechanism? Will the relationship between the annual report text's positive tone and green innovation be affected by the heterogeneity of economic policy uncertainty and industry attributes, and will it ultimately positively affect enterprise value? Although some scholars have discussed the relationship between the positive tone of management and green innovation [16], and found that the positive tone of management has a predictive effect and can significantly improve the level of green innovation of enterprises, they have not yet empirically explored the internal mechanism through which the positive tone of management affects the green innovation of enterprises, which provides space for our research. Based on this, we take Shanghai-Shenzhen A-share listed companies from 2010 to 2022 as samples and systematically discuss the impact of the annual report text's positive tone on enterprise green innovation and its mechanism to further expand and explain existing relevant studies. The empirical results show that the annual report text's positive tone plays a positive information incremental effect and significantly improves the enterprise's green innovation while improving the quantity and quality of green innovation. Through the mechanism test, we found that the positive tone of the annual report text can improve green innovation by easing corporate financing constraints and enhancing market attention. The results of the heterogeneity analysis show that the positive tone of the annual report text has a more significant effect on green innovation in industries with high economic policy uncertainty and non-heavily polluting industries. Finally, we found that the annual report text's positive tone is conducive to improving enterprise value.

The main contributions of this paper are as follows: Firstly, at present, the academic community mainly focuses on the impact of internal and external dimensions such as government policies, industry competition pressure, stakeholder pressure, gender diversity of the board of directors, and overseas experience of senior executives on corporate green innovation, while the research on the impact of incremental information in annual reports is still insufficient. This paper takes the annual report text's positive tone as the starting point, examines the relationship between the incremental information of the annual report text and green innovation, and complements the relevant research on the driving factors of green innovation. Secondly, this paper examines the relationship and mechanism between the annual report text's positive tone and corporate green innovation; we found that the annual report text's positive tone has a significant positive effect on corporate green innovation, providing empirical evidence for the "information increment view" of the corporate annual report text's positive tone. Thirdly, from the two dimensions of "quality" and "quantity," this paper divides green innovation into green innovation quantity and green innovation quality to identify whether the annual report text's positive tone has an "incremental" or "quality" impact on green innovation and provide some

references for in-depth research on relevant issues and market participants' investment decisions.

The remainder of this study is organized as follows: Section 2 reviews the extant literature, discusses the theoretical framework of hypotheses and the potential channel in detail. Section 3 outlines the definitions of data and variables, as well as the construction of models. Section 4 presents the results and discussion, addresses the issue of endogeneity and robustness tests, and investigates the potential channels. Section 5 conducts mechanism tests, heterogeneity analysis and economic effects tests. Section 6 summarizes the conclusions and discusses the insights, limitations and future perspectives. This paper's research flow chart is shown in Fig 1.

## 2. Literature review, theoretical analysis and hypothesis

Unlike financial information, the language expression of text information, especially the positive or negative tone, plays a particularly prominent role in conveying emotions and indicating attitudes [17]. The positive degree of an enterprise's tone of voice in information disclosure is

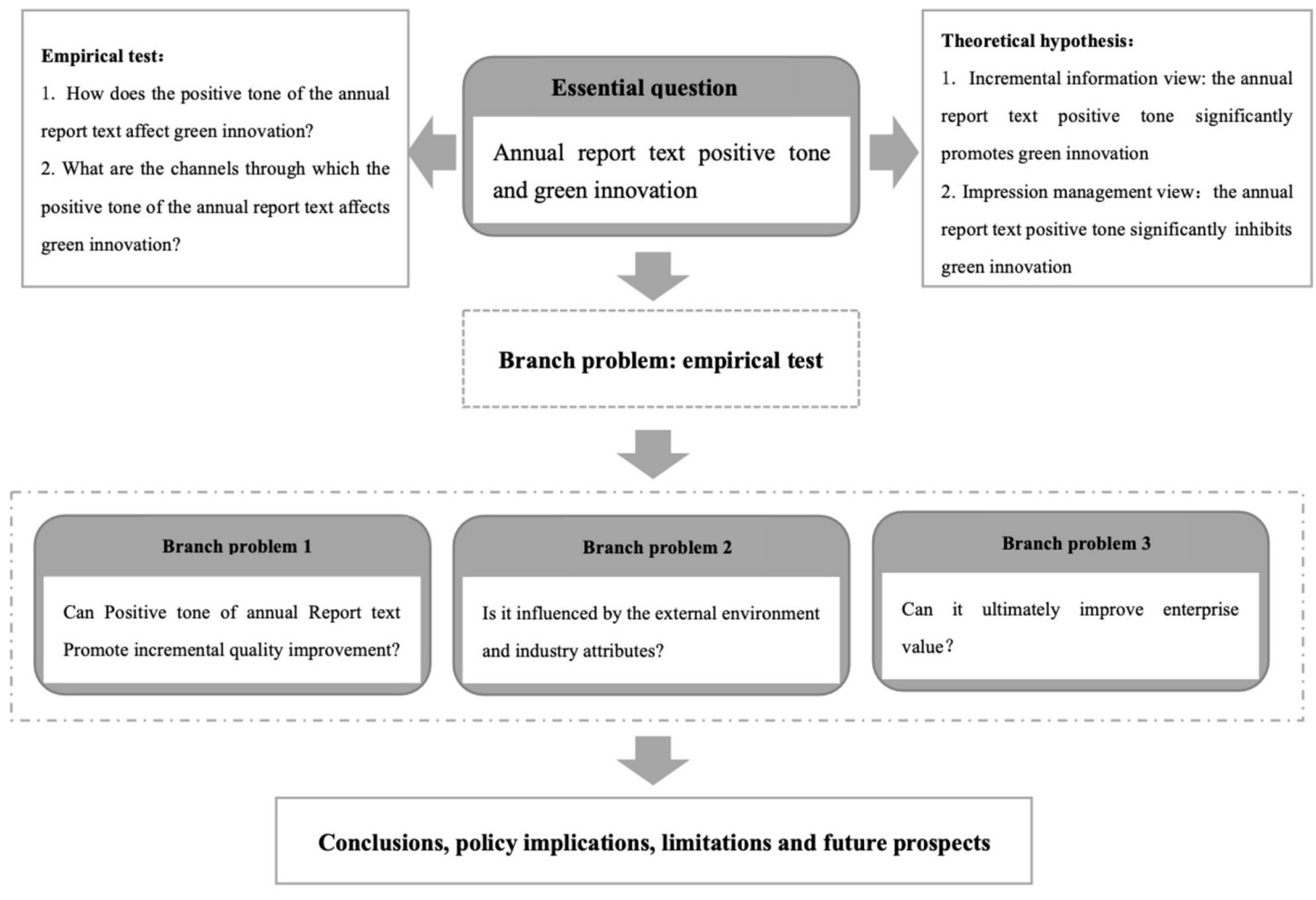

**Fig 1. The research flow chart.**

jointly affected by the enterprise's development status and the management's motivation and intention [18]. As for the intention and function of intonation in annual reports, the existing researches mainly include two opposing views: information increment view and impression management view. According to the information increment view, the intonation of the annual report text plays a fundamental incremental role. Managers can exert the signal transmission effect through positive intonation to convey the future performance of enterprises [11], stock market returns [12], and the performance of social responsibilities to capital market [13], increase the investment intensity of external investors [14] to effectively alleviate enterprises' financing constraints [15]. According to the impression management view, when the enterprise's business performance is poor, and the stock price needs to be increased, managers will have a solid motivation to mislead investors using intonation manipulation [18]. Even to meet the pro-environment behavior of investors, the management will convey the enterprise's green innovation strategy to the capital market through positive intonation [13]. Therefore, this paper analyzes the relationship and internal logic between the annual report text's positive tone and corporate green innovation from the information increment and impression management perspectives.

## 2.1 Information incremental view

Green innovation, as a combination of the two new development concepts of "green" and "innovation," has been widely recognized and supported by the public [19, 20]. However, compared with general technological innovation, green innovation has the characteristics of high risk, uncertainty and long-term periodicity, which will have a superimposed impact on the business risk of enterprises, and may eventually lead to the lack of internal motivation for green innovation [21, 22]. The positive tone of the management in the textual information disclosure can not only convey the good economic situation of the enterprise, but also illustrate the optimistic and risk-taking attitude of the manager. Such a positive and optimistic attitude is more likely to transform prosocial cognition into prosocial behaviors, such as charitable donation and environmental protection [23], to actively carry out green innovation activities. In addition, managers' risk preference will also more intuitively affect the structure and efficiency of innovation resources allocation, thus interfering with the willingness and behavior of enterprises to implement green innovation [24]. Managers who dare to take risks are more willing to make aggressive and risky strategic decisions and increase investment in green innovation to obtain high returns [25]. Secondly, innovation theory holds that available capital is vital in technological innovation. The natural high risk, uncertainty, and long-term cyclical characteristics of green innovation, as well as the existing knowledge spillover and technology spillover effect, also determine that sufficient capital is an essential guarantee for enterprises to carry out green innovation strategies, that is, financing constraints are a necessary obstacle for enterprises to implement green innovation strategies [26]. The positive intonation of managers in information disclosure can exert the signal transmission effect and convey to the outside world the company's future performance [11], stock market returns [12], and the implementation of social responsibilities [13], which can effectively alleviate the information asymmetry with investors. This enables investors to have a deeper understanding of the economic conditions of enterprises and increase their investment in enterprises [14, 27], to reduce financing constraints [15, 28], effectively solve the financing dilemma of enterprises' green innovation, and improve the willingness of enterprises to green innovation. Finally, the corporate annual report text's positive tone will attract the attention of external investors and be highly concerned by external supervision organizations such as analysts [29]. With the increase of attention, the probability and diffusion effect of the exposure of corporate green innovation will

also be greatly improved. In the current wave of green and low-carbon development, the public's environmental demands for green and low-carbon, ecological, and environmental protection are important driving forces for enterprises to carry out green innovation and development and achieve green transformation [28]. Therefore, under the supervision and pressure of external analysts, enterprises will actively implement green innovation activities to meet the demands of the public [6, 28]. Based on the above theoretical analysis, we propose the following hypotheses:

**H1a:** Based on the information increment view, the annual report text's positive tone can promote corporate green innovation

## 2.2 Impression management view

The impression management view holds that the management has a solid self-serving motive to manipulate the intonation by using the text information of the annual report [18]. As the regulatory authorities strengthen the supervision and punishment of digital information disclosure, the cost of manipulation by enterprise managers using digital information disclosure also increases, and low-cost intonation manipulation of text information has become a new means for management to manipulate information [30]. In particular, China's capital market is in the transition period, and there are serious governance problems such as information opacity and insufficient supervision, which provide a hotbed for the management to carry out strategic disclosure with the help of language expression. Following this logic, the annual report text's positive tone may not signal the enterprise's good economic performance and the manager's optimistic attitude; it is more likely to present a positive image of actively responding to the national green innovation development strategy. Managers use the impression management tool of intonation manipulation in information disclosure to cover up the poor performance of the enterprise in the environment and society. Specifically, managers will convey the enterprise's green innovation strategy to the capital market through a positive tone to mislead investors [13]. Existing studies have pointed out that the impression management concept is more suitable for the disclosure of listed companies in China, and the management is more likely to use the impression management concept for information disclosure, thus reducing the information efficiency of the capital market [31]. Based on the above theoretical analysis, we propose the following hypotheses:

**H1b**: Based on the impression management view, the annual report text's positive tone will inhibit corporate green innovation

## 3. Research design

### 3.1 Sample and data source

Considering that the concept of green development has been gradually strengthened since the Tenth National Congress of the Communist Party of China (CPC), our research since 2010, and taking Chinese A-share listed companies in Shenzhen and Shanghai from 2010 to 2022 as a sample. Following prior studies [16, 32], we excluded financial firms (e.g., banks, insurance firms, and investment trusts) with unique regulatory oversight and firms with special treatment (ST and *ST firms). We also eliminated the sample of companies with missing data. Finally, we are left with 34,830 observations. The continuous variables are winsorized at the 1 and 99 percentiles to alleviate the effect of outliers. In addition, we also adopt robust standard error to alleviate the heteroscedasticity problem.

We used Python to download the annual reports of Shanghai and Shenzhen A-share listed companies from 2010 to 2022 from www.cninfo.com.cn. The tone of the annual reports is from the dictionary data [17]. Green patent data were obtained from the China Research Data

Service Platform (CNRDS) database, and other data were obtained from the China Stock Market Accounting Research (CSMAR) database.

## 3.2 Variable definition

**3.2.1 The annual report text's positive tone.** The annual report text's positive tone (*Tone*). First of all, we take positive color words ("positive, "high quality," "promotion," etc.) and negative color words ("low," "unfavorable," "downward," etc.) as a list of positive and negative words respectively [11]. Then, we used Python programming to count the positive and negative words in the MD&A of the annual report to obtain the total number of positive and negative words [11]. Finally, we use Eq (1) to calculate the intonation positivity of the annual report text [33]. The larger the value of *Tone*, the more positive the report's tone.

$$Tone = \frac{\text{Total Positive words} - \text{Total negative words}}{\text{Total words}} \tag{1}$$

**3.2.2 Green innovation.** Green innovation (*Green*). Green patents are the most direct way to measure enterprises' green innovation degree [32]. Therefore, following prior studies [16, 32], we also take the number of green patent applications as its proxy variable. Among them, calculating the number of green patent applications covers the number of green inventions and practical patents. To overcome the right bias of green patent data, the value of the natural logarithm after the number of green patent applications +1 is finally used to measure.

**3.2.3 Control variables.** We control the following factors that may affect enterprises' green innovation: Corporate size (*Size*); Asset-liability ratio (*Lev*); Return on assets (*ROA*); Corporate growth (*Growth*); Cash flow (*Cashflow*); Listing age (*ListAge)*; Proportion of independent directors (*Indep*); The size of the Board (*Board*); Ownership concentration (*Top1*); Nature of property right (*SOE*); The CEO and chairman of the board are the same person (*Dual*). Other control variables: We also control the year effect (*Year*) and the industry effect (*Ind*). The specific definitions of variables are shown in Table 1.

## 3.3 Model setting

We construct the following regression model to investigate the relationship between the annual report text's positive tone and corporate green innovation.

$$Green_{i,t} = \alpha_0 + \alpha_1 \, Tone_{i,t} + \alpha_j \sum Controls_{i,t} + \sum Year_{i,t} + \sum Ind_{i,t} + \varepsilon_{i,t} \tag{2}$$

In Eq (2), *Green* means corporatṁe green innovation, and *Tone* means the annual report text's positive tone. The larger the value of *Tone*, the more positive the report's tone. If the regression coefficient $a_1$ in Eq (2) is significantly positive, indicating that the annual report text's positive tone plays an "information incremental effect", H1a is established. If the regression coefficient $a_1$ in Eq (2) is significantly negative, indicating that the annual report text's positive tone becomes a tool for enterprise management to carry out impression management, H1b is established.

# 4. Empirical results and analysis

## 4.1 Descriptive statistics

Table 2 presents descriptive statistics for the main variable. Specifically, the minimum value of green innovation (*Green*) is 0, the median is 0, the maximum is 6.6161, the mean is 0.3290, and the standard deviation is 0.7535, indicating that the green innovation activities of listed

**Table 1. Variables definitions.**

| variable | Specific Definition |
|---|---|
| Annual report text's positive tone | |
| Tone | $Tone = \frac{Total\ Positive\ words - Total\ negative\ words}{Total\ words}$ |
| Corporate green innovation | |
| Green | Ln (green patent application number + 1) |
| Control variables | |
| Size | Natural log of total assets |
| Lev | Debt-to-equity ratio |
| Top1 | Percentage of shareholding of the largest shareholder |
| Board | Total number of directors |
| Indep | The ratio of independent directors to the total number of directors |
| Dual | If the chairman and CEO are the same, the value is 1. Otherwise, 0 |
| ROA | Return on assets |
| Lisage | Current year - year of listing |
| SOE | If the enterprise is a state-owned enterprise, the value is 1, otherwise, 0 |
| Growth | (Current Year's operating revenue - last year's operating revenue)/last year's operating revenue |
| Cashflow | Net cash flows from operating activities/total assets |
| Ind | Industry dummy variable |
| Year | Year dummy variable |

companies in China are still in the preliminary stage, the overall level is low, and the degree of green innovation varies greatly among enterprises. Control variables: compared with the prior literature, the overall distribution of control variables is within a reasonable range.

## 4.2 Correlation analysis

Pearson correlation analysis results for significant variables are reported in Fig 2. The results show that the correlation coefficient between *Tone* (annual report text's positive tone) and *Green* (corporate green innovation) is significantly positive at the level of 5%, which preliminarily proves H1a, that is, positive tone of annual report text can improve the degree of corporate green innovation. In addition, the correlation coefficients among other relevant variables are all less than 0.5, meaning there is no severe multicollinearity problem among the relevant variables.

## 4.3 Regression result analysis

We use Eq (2) for regression to empirically investigate the relationship between the annual report text's positive tone and corporate green innovation. The regression results are reported in Table 3. Column (1) shows the regression results when only the fixed effect of year and industry is controlled, but other control variables that may affect green innovation are not included. It can be seen that the regression coefficient of *Tone* is 3.0803, significantly positive at the 1% level, indicating that the annual report text's positive tone can significantly improve the degree of green innovation. Column (2) shows the regression results when other control variables that may affect green innovation are further added, and the regression coefficient of *Tone* is 1.6718, which is significant at the 1% level. The empirical results show that the annual report text's positive tone plays an "information incremental effect" [11, 12, 28], and positively

**Table 2. Variable descriptive statistics.**

| VarName | Obs | Mean | SD | Min | Median | Max |
| --- | --- | --- | --- | --- | --- | --- |
| Green | 34830 | 0.3290 | 0.7535 | 0.0000 | 0.0000 | 6.6161 |
| Tone | 34830 | 0.0345 | 0.0160 | -0.2000 | 0.0350 | 0.1484 |
| Size | 34830 | 22.2214 | 1.2870 | 19.5850 | 22.0369 | 26.4523 |
| Lev | 34830 | 0.4258 | 0.2050 | 0.0274 | 0.4187 | 0.9079 |
| ROA | 34830 | 0.0399 | 0.0656 | -0.3730 | 0.0385 | 0.2473 |
| Cashflow | 34830 | 0.0461 | 0.0693 | -0.2218 | 0.0453 | 0.2669 |
| Growth | 34830 | 0.1703 | 0.4163 | -0.6576 | 0.1066 | 4.0242 |
| Board | 34830 | 2.1208 | 0.1973 | 1.6094 | 2.1972 | 2.7081 |
| Indep | 34830 | 37.6096 | 5.3763 | 27.2700 | 36.3600 | 60.0000 |
| Top1 | 34830 | 34.3295 | 14.6872 | 8.0204 | 32.0854 | 75.8433 |
| ListAge | 34830 | 2.1434 | 0.8272 | 0.0000 | 2.3026 | 3.4012 |
| Dual | 34830 | 0.2817 | 0.4498 | 0.0000 | 0.0000 | 1.0000 |
| SOE | 34830 | 0.3576 | 0.4793 | 0.0000 | 0.0000 | 1.0000 |

promotes the green innovation behavior of enterprises, that is, H1a was verified. This conclusion is consistent with the relevant research conclusion [16].

## 4.4 Can the annual report text's positive tone improve the quantity and quality of green innovation?

Existing studies have pointed out that under multiple pressures, China's listed companies have a severe phenomenon of "heavy quantity, light quality" [34]. To further investigate whether the positive tone of the annual report text plays an "incremental" or "qualitative" role in green innovation? We use Green_1, a proxy indicator of green innovation quality, as the value +1 after the natural logarithm of green invention patent applications [34]. we use Green_2 a proxy indicator of green innovation quantity, as the value of the natural logarithm +1 of the number of green utility model patent applications [34]. Table 4 reports the regression results. The results show that the annual report text's positive tone has significantly improved enterprises' green innovation from both the quantitative and quality.

## 4.5 Robustness test

**4.5.1 Replace the annual report text's positive tone.** To overcome the influence of the measurement bias of explanatory variables, first of all, we take positive color words (" positive, "high quality," "promotion," etc.) and negative color words ("low, "unfavorable," "downward," etc.) as a list of positive and negative words respectively [11]. Then, we used Python programming to count the positive and negative words in the MD&A of the annual report to obtain the total number of positive and negative words. Finally, we use the Eq (3) to calculate the report text's positive tone (Tone2) [33]. The regression results are shown in column (1) of Table 5. The regression coefficient of Tone2 is significantly positive at a 1% level, indicating that our conclusion is relatively robust.

$$Tone2 = \frac{(\text{Total Positive words} - \text{Total negative words})}{\text{Total Positive words} + \text{Total negative words}} \tag{3}$$

**4.5.2 Individual fixed effect.** We retest the principal regression using a fixed effects model for controlling individuals to exclude the endogenous effects of missing variables that

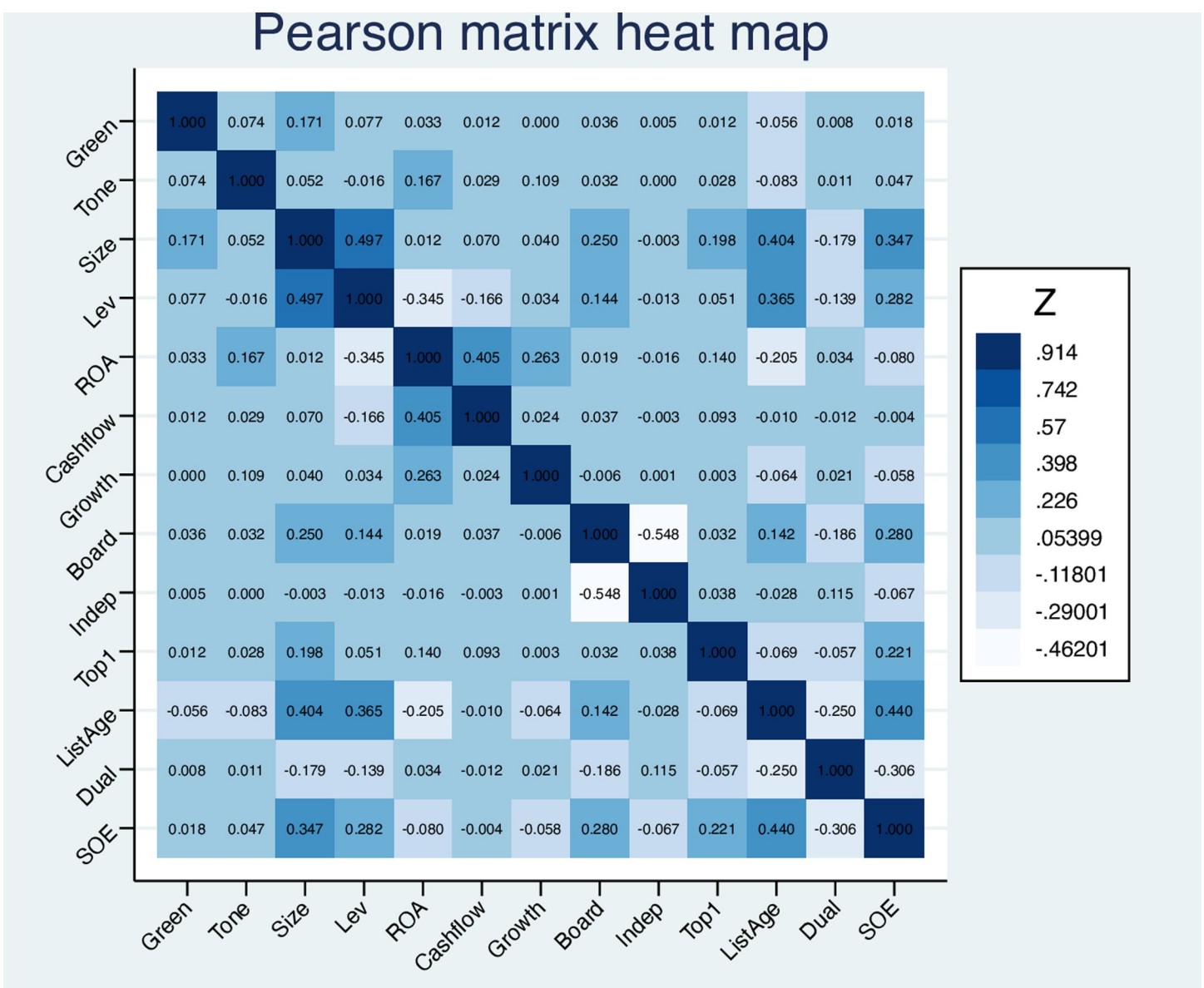

**Fig 2. Correlation analysis heat map.**

are unobservable to the firm and do not change over time. As shown in column (2) of Table 5, the regression coefficient of *Tone* is significantly positive at a 5% level, and the main research results of this paper remain unchanged.

**4.5.3 The positive tone of the annual report text lags one period.** Considering the possible problem of reverse causal endogeneity in the research, enterprises with a high degree of green innovation may increase the annual report text's positive tone. To avoid the interference of this endogenous problem, the annual report text's positive tone and all control variables are delayed for one period, and we re-regression Eq (2). The results reported in column (3) of Table 5 show that the regression coefficient of *Tone1* with a one-stage lag is significantly positive at a 1% level, and our main findings remain unchanged.

**Table 3. The annual report text's positive tone and corporate green innovation.**

| Variable | Green | |
|---|---|---|
| | **(1)** | **(2)** |
| *Tone* | 3.0803*** | 1.6718*** |
| | (12.4728) | (6.8097) |
| *Size* | | 0.1364*** |
| | | (25.7814) |
| *Lev* | | 0.1354*** |
| | | (5.9476) |
| *ROA* | | 0.3396*** |
| | | (5.2719) |
| *Cashflow* | | 0.0919 |
| | | (1.5943) |
| *Growth* | | -0.0554*** |
| | | (-7.2372) |
| *Board* | | 0.0742*** |
| | | (2.6798) |
| *Indep* | | 0.0009 |
| | | (1.0442) |
| *Top1* | | -0.0005 |
| | | (-1.6190) |
| *ListAge* | | -0.0794*** |
| | | (-13.6288) |
| *Dual* | | 0.0171* |
| | | (1.9161) |
| *SOE* | | 0.0768*** |
| | | (7.5685) |
| _cons | -0.2208*** | -2.9477*** |
| | (-7.8117) | (-23.8754) |
| Year fe | Yes | Yes |
| Ind fe | Yes | Yes |
| N | 34830 | 34830 |
| r2 | 0.1275 | 0.1764 |

Note: ***p<0.01

**p<0.05

*p<0.1 are significant at the level of 1%, 5% and 10%, respectively. The result reported in parentheses is a t statistic.

**4.5.4 Replacement regression model.** Since the raw data of the number of green patent applications are mostly 0, to avoid data estimation bias, we select the original data of the number of green patent applications; that is, the natural logarithm of +1 is no longer processed. Then, we used the Tobit model, Poisson model, and Panel negative binomial model to test empirically. Column (4)-(6) of Table 5 shows that the regression coefficient of *Tone* is significantly positive at the 1% level, which still supports hypothesis H1a.

**4.5.5 Propensity score matching method.** The annual report text's positive tone and corporate green innovation may be affected by some corporate characteristics simultaneously, resulting in the endogenous problem that the conclusions of this paper are affected by sample bias. To solve this problem, we use the propensity score matching method (PSM) [35]. First, the samples were divided into experimental and control groups according to the median of the

**Table 4. The annual report text's positive tone and the quantity and quality of green innovation.**

| Variable | Green_1 | Green_2 |
|---|---|---|
| | (1) | (2) |
| Tone | 1.3540*** | 0.6492*** |
| | (6.6014) | (3.8233) |
| Size | 0.1151*** | 0.0794*** |
| | (24.7250) | (20.7012) |
| Lev | 0.0711*** | 0.0972*** |
| | (3.8473) | (5.9179) |
| ROA | 0.2760*** | 0.1449*** |
| | (5.2454) | (3.1708) |
| Cashflow | 0.0551 | 0.0758* |
| | (1.1723) | (1.8086) |
| Growth | -0.0445*** | -0.0265*** |
| | (-7.3634) | (-4.7368) |
| Board | 0.0472** | 0.0622*** |
| | (1.9999) | (3.1875) |
| Indep | 0.0009 | 0.0009 |
| | (1.2491) | (1.3778) |
| Top1 | -0.0006** | -0.0002 |
| | (-2.3750) | (-0.8068) |
| ListAge | -0.0520*** | -0.0582*** |
| | (-10.9796) | (-13.6662) |
| Dual | 0.0208*** | 0.0144** |
| | (2.8342) | (2.1543) |
| SOE | 0.0778*** | 0.0270*** |
| | (9.1623) | (3.7640) |
| _cons | -2.4583*** | -1.7453*** |
| | (-22.8268) | (-20.0710) |
| Year fe | Yes | Yes |
| Ind fe | Yes | Yes |
| N | 34830 | 34830 |
| r2 | 0.1516 | 0.1500 |

Note: ***p<0.01

**p<0.05

*p<0.1 are significant at the level of 1%, 5% and 10%, respectively. The result reported in parentheses is a t statistic.

annual report text's positive tone. The sample companies whose annual report text's positive tone is greater than the median value is set as the experimental group, the value is 1, and the other sample companies are set as the control group, the value is 0. According to *ROA*, *Growth*, *SOE*, *ListAge*, *Size*, *Top1*, *Cashflow*, *Dual*, *Lev*, we use the nearest neighbor propensity score to identify matching samples and perform a balanced test on matching samples. Table 6 shows that after the matching, the standardized errors are all less than 10%, which proves that the matching is successful. Finally, we use Eq 2 to regress Hypothesis 1 on the matched samples. The regression results after matching are reported in column (7) of Table 5. The regression coefficient of *Tone* (1.4924) passed the positive significance test at a 1% level. That is, after considering the endogeneity caused by sample bias, the results are consistent with the results of the primary regression test.

**Table 5. Robust test.**

| Variable | Green | | | | | | |
|---|---|---|---|---|---|---|---|
| | (1) | (2) | (3) | (4) | (5) | (6) | (7) |
| Tone2 | 0.2194*** | | | | | | |
| | (6.6538) | | | | | | |
| Tonel | | | 1.6432*** | | | | |
| | | | (6.0218) | | | | |
| Tone | | 0.6444** | | 6.1536*** | 2.0151*** | 2.9596*** | 1.4924*** |
| | | (2.3653) | | (4.2626) | (4.0660) | (3.2949) | (4.5248) |
| Size | 0.1353*** | 0.0664*** | 0.1391*** | 0.8023*** | 0.2661*** | 0.2934*** | 0.1402*** |
| | (25.4433) | (7.1898) | (23.5238) | (34.1684) | (16.6515) | (14.9170) | (19.6426) |
| Lev | 0.1369*** | 0.0703** | 0.1438*** | 0.4349*** | 0.0592 | 0.1959* | 0.1193*** |
| | (6.0137) | (2.2412) | (5.5946) | (3.0207) | (0.8190) | (1.7978) | (3.8396) |
| ROA | 0.3326*** | 0.1091* | 0.5310*** | 1.7136*** | 0.9329*** | 0.8360*** | 0.3814*** |
| | (5.1579) | (1.8698) | (7.2333) | (4.0731) | (6.2907) | (3.2283) | (4.0698) |
| Cashflow | 0.0972* | -0.1186** | 0.1861*** | 0.4241 | -0.6545*** | -0.2769 | 0.0912 |
| | (1.6872) | (-2.5189) | (2.9636) | (1.1964) | (-6.0316) | (-1.3480) | (1.1245) |
| Growth | -0.0567*** | -0.0247*** | -0.0400*** | -0.2596*** | -0.1429*** | -0.1532*** | -0.0568*** |
| | (-7.3876) | (-4.7950) | (-4.8161) | (-4.7045) | (-7.1973) | (-4.3851) | (-5.6262) |
| Board | 0.0745*** | -0.0190 | 0.0900*** | 0.3437** | 0.0504 | 0.0811 | 0.0954** |
| | (2.6923) | (-0.4565) | (2.9415) | (2.4098) | (0.9062) | (0.8477) | (2.5545) |
| Indep | 0.0009 | 0.0006 | 0.0008 | 0.0075 | 0.0026 | 0.0030 | 0.0015 |
| | (1.0483) | (0.4663) | (0.8324) | (1.5427) | (1.5733) | (1.0042) | (1.2912) |
| Top1 | -0.0004 | -0.0010** | -0.0007** | -0.0033** | 0.0007 | -0.0003 | -0.0005 |
| | (-1.5175) | (-1.9653) | (-2.0521) | (-2.0460) | (0.7100) | (-0.2549) | (-1.2760) |
| ListAge | -0.0781*** | -0.0634*** | -0.0818*** | -0.3522*** | -0.1819*** | -0.2974*** | -0.0835*** |
| | (-13.3105) | (-6.4202) | (-12.6850) | (-10.3956) | (-8.2623) | (-11.0352) | (-10.4268) |
| Dual | 0.0167* | -0.0042 | 0.0233** | 0.1418*** | 0.0115 | 0.0399 | 0.0250** |
| | (1.8610) | (-0.4062) | (2.3523) | (2.7698) | (0.5760) | (1.1957) | (2.0117) |
| SOE | 0.0795*** | 0.0267 | 0.0891*** | 0.3621*** | 0.0599* | 0.2585*** | 0.0821*** |
| | (7.8369) | (1.3367) | (7.9759) | (6.2299) | (1.7590) | (5.4510) | (5.9843) |
| _cons | -2.9355*** | -1.1219*** | -2.8572*** | -18.5460*** | -7.7533*** | -9.1840*** | -3.2375*** |
| | (-23.7648) | (-5.2238) | (-7.9811) | (-29.5638) | (-15.9446) | (-13.3683) | (-17.2312) |
| Year fe | Yes | Yes | Yes | Yes | Yes | Yes | Yes |
| Ind fe | Yes | Yes | Yes | Yes | Yes | Yes | Yes |
| N | 34830 | 34830 | 29824 | 34830 | 34830 | 34830 | 18778 |
| r2/Pseudo R2 | 0.1763 | 0.1756 | 0.1802 | 0.1759 | 0.1850 | 0.1766 | 0.1852 |

Note: ***p<0.01

**p<0.05

*p<0.1 are significant at the level of 1%, 5% and 10%, respectively. The result reported in parentheses is a t statistic.

**4.5.6 Instrumental variable method.** Considering that there may be a reverse causal relationship between the positive tone of annual report texts and green innovation, we select the mean value of the positive tone of texts in the same year and industry as the instrumental variable (*Tone_m*) and adopt the two-stage least squares method (IV-2SLS) [33]. We adopt this idea of instrumental variables mainly for the following two reasons: Firstly, enterprises within the same industry have similar operation and business scope, and will be affected by the same market environment and regulatory rules. Therefore, the narrative logic and expression of the

**Table 6. Balance test.**

| Variable | Unmatched Matched | Mean Treated Control | | %bias | %reduct \|bias\| | t-test | |
|---|---|---|---|---|---|---|---|
| | | | | | | t | p>\|t\| |
| ROA | U | .04824 | .03175 | 25.4 | 99.9 | 23.65 | 0.000 |
| | M | .04824 | .04822 | 0.0 | | 0.02 | 0.982 |
| Growth | U | .20704 | .13405 | 17.6 | 87.2 | 16.42 | 0.000 |
| | M | .20704 | .19773 | 2.2 | | 2.01 | 0.045 |
| SOE | U | .37334 | .34203 | 6.5 | 79.7 | 6.10 | 0.000 |
| | M | .37334 | .37971 | -1.3 | | -1.22 | 0.222 |
| ListAge | U | 2.0878 | 2.1981 | -13.4 | 97.4 | -12.48 | 0.000 |
| | M | 2.0878 | 2.0907 | -0.3 | | -0.32 | 0.751 |
| Size | U | 22.263 | 22.18 | 6.4 | 85.4 | 6.00 | 0.000 |
| | M | 22.263 | 22.275 | -0.9 | | -0.86 | 0.390 |
| Top1 | U | 34.464 | 34.196 | 1.8 | 66.5 | 1.70 | 0.088 |
| | M | 34.464 | 34.554 | -0.6 | | -0.57 | 0.570 |
| Cashflow | U | .04714 | .04502 | 3.1 | 65.3 | 2.86 | 0.004 |
| | M | .04714 | .04641 | 1.1 | | 0.98 | 0.329 |
| Dual | U | .28785 | .27552 | 2.7 | 87.8 | 2.56 | 0.011 |
| | M | .28785 | .28935 | -0.3 | | -0.31 | 0.758 |
| Lev | U | .42338 | .42829 | -2.4 | 68.5 | -2.23 | 0.026 |
| | M | .42338 | .42184 | 0.8 | | 0.71 | 0.480 |

annual report texts of enterprises in the same industry tend to be more similar, presenting certain industry characteristics, which meets the relevance requirements of the instrumental variables. Secondly, the language characteristics of other enterprises' annual report texts do not have an impact on this enterprise's green innovation, which meets the requirement of exogeneity of instrumental variables.

The regression results are shown in Table 7. The test results of weak instrumental variables show that the F value is 3054.94, which is significantly greater than the empirical value 10. The instrumental variables satisfy the correlation and are strong instrumental variables. The regression coefficient of *Tone* is still significantly positive, which once again verifies the positive tone of the annual report text to promote enterprises' green innovation.

**4.5.7 Adding macro variables.** To make the results more robust, we added a macroeconomic variable, the value-added rate of gross domestic product (*GDP*), as a control variable. The regression results after adding macroeconomic variables are shown in Table 8. The regression coefficient of *Tone* is 1.6409, which is significant at the 1% level, indicating that our main regression results are still robust.

## 5. Extended analysis

### 5.1 Mechanism analysis

The above empirical results support the "information increment view" hypothesis of the positive tone of annual report text; that is, the positive tone of annual report text significantly improves the degree of green innovation, but it does not answer the question of how positive tone of annual report text affects corporate green innovation. The theoretical analysis points out that under the "information increment view," on the one hand, the positive tone of the annual report text can release more positive signals about enterprises and effectively alleviate the financing constraints. Reducing financing constraints can effectively help the financing difficulties of enterprises' green innovation and improve their willingness to green R&D and innovation. On the other hand, the positive tone of the annual report text will significantly

**Table 7. Instrumental variable method.**

| Variable | Green | |
|---|---|---|
| | **(1)** | **(2)** |
| Tone_m | 1.3852** | |
| | (2.0597) | |
| Tone | | 1.7996** |
| | | (2.0609) |
| Size | 0.1372*** | 0.1364*** |
| | (33.8924) | (33.2899) |
| Lev | 0.1407*** | 0.1352*** |
| | (5.6741) | (5.4336) |
| ROA | 0.4021*** | 0.3346*** |
| | (5.5959) | (4.2046) |
| Cashflow | 0.0840 | 0.0920 |
| | (1.3763) | (1.5049) |
| Growth | -0.0509*** | -0.0554*** |
| | (-5.3560) | (-5.6942) |
| Board | 0.0765*** | 0.0742*** |
| | (3.1126) | (3.0151) |
| Indep | 0.0009 | 0.0009 |
| | (1.1232) | (1.1035) |
| Top1 | -0.0005* | -0.0005* |
| | (-1.7100) | (-1.6592) |
| ListAge | -0.0812*** | -0.0792*** |
| | (-13.9104) | (-13.2109) |
| Dual | 0.0177** | 0.0171* |
| | (2.0134) | (1.9424) |
| SOE | 0.0814*** | 0.0763*** |
| | (8.1450) | (7.2863) |
| _cons | -2.9496*** | -2.9495*** |
| | (-9.2057) | (-9.2108) |
| Year fe | Yes | Yes |
| Ind fe | Yes | Yes |
| F-value | 3054.94 | |
| N | 34799 | 34799 |
| r2 | 0.1754 | 0.1764 |

Note: ***p<0.01

**p<0.05

*p<0.1 are significant at the level of 1%, 5% and 10%, respectively. The result reported in parentheses is a t statistic.

enhance the external attention of the company. Under the pressure of external awareness, enterprises will actively implement green innovation activities to meet stakeholders' demands. To verify whether the above logic is valid, we replace the explained variables in Eq (2) with the financing constraint *WW* and the analyst attention *Ans* (the number of analysts tracking +1 is the natural logarithm), and conducts a regression again [36].

Table 9 reports the mechanism analysis results. Specifically, the regression coefficient of *Tone* in column (1) is -0.0707, which is significant at the 1% level, indicating that the positive tone of the annual report text significantly reduces enterprises' financing constraint (*WW*). In

**Table 8. Robust test after adding macro variables.**

| Variable | Green |
|---|---|
| | (1) |
| Tone | 1.6409*** |
| | (6.6681) |
| Size | 0.1360*** |
| | (25.5848) |
| Lev | 0.1385*** |
| | (6.0557) |
| ROA | 0.3509*** |
| | (5.4223) |
| Cashflow | 0.0925 |
| | (1.5993) |
| Growth | -0.0562*** |
| | (-7.3296) |
| Board | 0.0703** |
| | (2.5238) |
| Indep | 0.0008 |
| | (0.9061) |
| Top1 | -0.0005* |
| | (-1.7102) |
| ListAge | -0.0813*** |
| | (-13.7965) |
| Dual | 0.0166* |
| | (1.8508) |
| SOE | 0.0789*** |
| | (7.7456) |
| GDP | -0.0160*** |
| | (-6.4435) |
| _cons | -2.7560*** |
| | (-21.4502) |
| Year fe | Yes |
| Ind fe | Yes |
| N | 34664 |
| r2 | 0.1769 |

Note: ***$p<0.01$

**$p<0.05$

*$p<0.1$ are significant at the level of 1%, 5% and 10%, respectively. The result reported in parentheses is a t statistic.

column (2), the regression coefficient of analyst attention (*Ans*) is 3.2408, which is significant at the 1% level, indicating that the positive tone of the annual report text can significantly improve analyst attention. To ensure the reliability of test results, we conducted Sobel test, z value was 6.473 and 7.374. The mechanism results show that the annual report text's positive tone can positively affect enterprises' green innovation through the two ways of alleviating financing constraints and enhancing external attention.

**Table 9. Mechanism analysis.**

| Variable | WW | Ans |
|---|---|---|
| | (1) | (2) |
| Tone | -0.0707*** | 3.2408*** |
| | (-6.3821) | (10.1206) |
| Size | -0.0485*** | 0.5465*** |
| | (-2.6e+02) | (107.0802) |
| Lev | 0.0243*** | -0.4401*** |
| | (21.0272) | (-13.5377) |
| ROA | -0.1411*** | 4.9827*** |
| | (-40.3301) | (43.2836) |
| Cashflow | -0.1033*** | 0.7663*** |
| | (-36.4017) | (9.3624) |
| Growth | -0.0418*** | 0.0414*** |
| | (-57.1731) | (2.9598) |
| Board | -0.0029*** | 0.1346*** |
| | (-2.6948) | (4.1986) |
| Indep | 0.0000 | 0.0012 |
| | (0.4025) | (1.1165) |
| Top1 | -0.0001*** | -0.0041*** |
| | (-7.1335) | (-11.1922) |
| ListAge | 0.0058*** | -0.2052*** |
| | (21.9542) | (-26.1280) |
| Dual | -0.0003 | 0.0907*** |
| | (-0.6674) | (7.7835) |
| SOE | -0.0013*** | -0.2478*** |
| | (-2.9370) | (-18.7217) |
| _cons | 0.2281*** | -10.0192*** |
| | (52.4321) | (-79.9368) |
| Year fe | Yes | Yes |
| Ind fe | Yes | Yes |
| Sobel-Z | 6.473 | 7.374 |
| N | 30225 | 33650 |
| r2 | 0.8561 | 0.4538 |

Note: ***p<0.01

**p<0.05

*p<0.1 are significant at the level of 1%, 5% and 10%, respectively. The result reported in parentheses is a t statistic.

## 5.2 Heterogeneity analysis

**5.2.1 The influence of economic policy uncertainty.** China, transitioning from high-speed economic growth to high-quality development, faces significantly more risks and challenges, and its economic development has significant destabilizing and uncertain factors. To cope with the downward economic pressure brought about by risks and challenges, China has continuously carried out macroeconomic policy reform and innovation, and the uncertainty of economic policy has increased significantly, which not only directly affects the efficiency and effect of the macroeconomic operation, but also spreads to micro-enterprises through the capital market mechanism. Finally, economic policy uncertainty will become an important external force affecting the green innovation of micro-enterprises. The rise of economic policy

uncertainty will exacerbate market information asymmetry and increase enterprises' financing difficulty, thus inhibiting enterprises' green innovation [37, 38]. The positive tone of the annual report can reduce the information asymmetry between enterprises and external markets, thus easing the financing constraints of enterprises and providing funds for enterprises to carry out green innovation. Therefore, we predict that the positive tone of the annual report text will promote corporate green innovation more significantly in listed companies with high economic policy uncertainty.

we adopted the uncertainty index of China's economic policy based on the South China Morning Post, a news report retrieval platform, through keyword search [39, 40]. Then, the logarithm of the arithmetical mean of the economic policy uncertainty index for each month of the year is further calculated, and the logarithm of the monthly indicator is converted into the logarithm of the annual indicator and then divided by 100 for standardization processing as the final proxy indicator (*EPU*) of economic policy uncertainty.

We use the mean value of the economic policy uncertainty index of listed companies as the grouping criterion. If the economic policy uncertainty faced by enterprises is lower than the mean value of the index, they are divided into the low economic policy uncertainty group; if the economic policy uncertainty is higher than the mean value of the index, they are divided into the high economic policy uncertainty group. If the positive tone coefficient of the annual report is significantly different in different subgroup samples, it indicates that the positive tone of the annual report has different effects on the green innovation of enterprises under different external environments.

The regression results are shown in Table 10. When economic policy uncertainty is high, the coefficient of *Tone* is 2.5714, which is significant at the 1% level. When economic policy uncertainty is low, the coefficient of *Tone* is 0.9124, also significant at the 1% level. However, when we further used the Bootstrap sampling method to test the significance of inter-group coefficient differences, the corresponding P-value was 0.000, indicating that when the economic policy uncertainty was higher, the positive tone of the annual report had a more obvious role in promoting corporate green innovation. This test result further confirmed the information incremental effect of positive tone of the text.

**5.2.2 The influence of industry attributes.** Industry heterogeneity will lead to differences in corporate ethics and environmental awareness [41], which may influence enterprises' choices of green innovation strategy. Therefore, industry heterogeneity may significantly affect the relationship between the positive tone of the annual report text and corporate green innovation. Compared with non-heavy polluting industries, heavy polluting industries will be subject to more stringent environmental supervision and pressure from environmental protection policies. On the other hand, green innovation will bring enterprises a better reputation and performance [42], so the heavily polluting industries have a stronger incentive to implement green innovation. While non-heavy polluters often do not have a political advantage, when companies use a positive tone, they need to attract the attention of stakeholders and maintain a good corporate reputation for green innovation. Therefore, we predict that the positive relationship between positive text tone and green innovation is more obvious in non-heavy polluting enterprises.

According to the industry classification management directory of environmental protection verification of listed companies, we divided the sample enterprises into two sub-samples of heavy pollution enterprises and non-heavy pollution enterprises for heterogeneity test. The results in Table 11 show that the positive tone of the annual report text in non-heavy polluting industries is significantly and positively correlated with green innovation. In contrast, in heavy-polluting industries, *Tone*'s regression coefficient is positive but insignificant. When we further used the Bootstrap sampling method to test the significance of coefficient differences

**Table 10. The influence of economic policy uncertainty.**

| Variable | Green | |
|---|---|---|
| | (1)<br>EPU_H | (2)<br>EPU_L |
| Tone | 2.5714*** | 0.9124*** |
| | (6.2380) | (3.0190) |
| Size | 0.1471*** | 0.1316*** |
| | (17.1074) | (19.4592) |
| Lev | 0.1554*** | 0.1105*** |
| | (4.1084) | (3.8429) |
| ROA | 0.2113** | 0.4864*** |
| | (2.1558) | (5.2977) |
| Cashflow | 0.1457 | 0.0511 |
| | (1.3894) | (0.7594) |
| Growth | -0.0578*** | -0.0565*** |
| | (-3.1368) | (-7.0944) |
| Board | 0.0502 | 0.0921*** |
| | (1.1079) | (2.6060) |
| Indep | 0.0007 | 0.0010 |
| | (0.4558) | (0.9090) |
| Top1 | -0.0002 | -0.0006* |
| | (-0.3215) | (-1.7762) |
| ListAge | -0.0947*** | -0.0699*** |
| | (-10.0420) | (-9.4920) |
| Dual | 0.0119 | 0.0211* |
| | (0.8427) | (1.8371) |
| SOE | 0.0858*** | 0.0719*** |
| | (4.8841) | (5.8116) |
| _cons | -3.2101*** | -3.1034*** |
| | (-12.5828) | (-19.4685) |
| Year fe | Yes | Yes |
| Ind fe | Yes | Yes |
| p-value | 0.000 | |
| N | 14539 | 20291 |
| r2 | 0.1820 | 0.1737 |

Note: ***$p<0.01$

**$p<0.05$

*$p<0.1$ are significant at the level of 1%, 5% and 10%, respectively. The result reported in parentheses is a t statistic.

between groups again, the corresponding P value was 0.030, indicating that the positive impact of the positive tone of annual report text on enterprise green innovation was more significant in non-heavy pollution enterprises. This result further confirms the information increment effect of positive text intonation.

**5.2.3 Corporate value effect analysis.** Green innovation is a vital strategy implemented by enterprises, and whether the implementation of the strategy can ultimately enhance the value of enterprises is also the most concerned issue of enterprises. Therefore, this paper further examines whether the positive tone of the annual report text has a positive impact on the implementation of the green innovation strategy of enterprises that can effectively improve

**Table 11. The influence of industry attributes.**

| Variable | Green | |
|---|---|---|
| | **(1)** Heavy-polluting industries | **(2)** Non-heavy polluting industries |
| Tone | 0.9926** | 1.8604*** |
| | (2.2232) | (6.3929) |
| Size | 0.1251*** | 0.1388*** |
| | (15.3121) | (21.4571) |
| Lev | -0.1300*** | 0.2124*** |
| | (-2.8428) | (8.0826) |
| ROA | 0.0166 | 0.4077*** |
| | (0.1184) | (5.6243) |
| Cashflow | 0.3876*** | 0.0133 |
| | (3.4582) | (0.1992) |
| Growth | -0.0371** | -0.0586*** |
| | (-2.3156) | (-6.6983) |
| Board | 0.2627*** | 0.0264 |
| | (4.9257) | (0.8153) |
| Indep | 0.0048** | -0.0002 |
| | (2.5740) | (-0.2030) |
| Top1 | 0.0007 | -0.0007** |
| | (1.2346) | (-2.0890) |
| ListAge | -0.0736*** | -0.0818*** |
| | (-6.3576) | (-12.1831) |
| Dual | -0.0440*** | 0.0320*** |
| | (-2.7101) | (3.0687) |
| SOE | 0.1037*** | 0.0671*** |
| | (5.4438) | (5.6458) |
| _cons | -3.0160*** | -3.0908*** |
| | (-10.8181) | (-20.1513) |
| Year fe | Yes | Yes |
| Ind fe | Yes | Yes |
| p-value | 0.030 | |
| N | 7892 | 26938 |
| r2 | 0.1411 | 0.1861 |

Note: ***p<0.01

**p<0.05

*p<0.1 are significant at the level of 1%, 5% and 10%, respectively. The result reported in parentheses is a t statistic.

corporate value. Since enterprises have invested a lot in the early stage of implementing a green innovation strategy, its economic benefits may not be reflected in the short term. Therefore, for corporate value indicators, we adopt Tobin's Q ratio (*TobinQ*) and the Price-to-Book ratio (*PB*) to measure [28], focusing on the impact on the long-term value of enterprises, and construct the following Eq (4) to test the effect of positive tone of annual report text and green innovation on corporate value. The other control variables in Eq (4) are consistent with Eq (2), focusing on the coefficient of cross-term *Tone*Green*.

The regression results are shown in Table 12. In column (1) and (2), the coefficient of *Tone*Green* is significantly positive at a 1% level, indicating that the positive tone of the annual

**Table 12. Economic effect analysis.**

| Variable | TobinQ | PB |
|---|---|---|
|  | (1) | (2) |
| Green | 0.0659*** | 0.1346*** |
|  | (4.6635) | (6.6859) |
| Tone | -7.8866*** | -9.3508*** |
|  | (-3.5345) | (-8.0397) |
| Green* Tone | 3.1469*** | 3.6602*** |
|  | (4.2474) | (2.9340) |
| Size | -0.7457*** | -1.2969*** |
|  | (-20.1310) | (-48.8915) |
| Lev | 0.8744*** | 5.6605*** |
|  | (3.7399) | (35.7679) |
| ROA | 4.3115*** | 8.5022*** |
|  | (8.9011) | (19.1690) |
| Cashflow | -0.1414 | 1.8864*** |
|  | (-0.2729) | (6.1135) |
| Growth | 0.0204 | 0.4426*** |
|  | (0.4918) | (7.8507) |
| Board | -0.0531 | 0.3067*** |
|  | (-0.2684) | (2.9883) |
| Indep | 1.4093*** | 0.0247*** |
|  | (3.5678) | (7.4342) |
| Top1 | -0.1251 | 0.0057*** |
|  | (-1.5010) | (5.3486) |
| ListAge | 0.5605*** | 0.0907*** |
|  | (18.3238) | (3.5093) |
| Dual | 0.0363 | 0.0595* |
|  | (0.9028) | (1.6768) |
| SOE | -0.2137*** | -0.2957*** |
|  | (-5.2959) | (-7.4701) |
| _cons | 16.7769*** | 26.0452*** |
|  | (16.8289) | (47.8295) |
| Year fe | Yes | Yes |
| Ind fe | Yes | Yes |
| N | 29190 | 34288 |
| r2 | 0.1375 | 0.3316 |

Note: ***p<0.01

**p<0.05

*p<0.1 are significant at the level of 1%, 5% and 10%, respectively. The result reported in parentheses is a t statistic.

report text has a positive impact on corporate green innovation and can improve corporate value.

## 6. Conclusion

In the context of China's comprehensive implementation of the stock issuance registration system reform with information disclosure as the core and the promotion of the coordinated development of green and innovation, using the data of Shanghai and Shenzhen A-share listed

Chinese companies from 2010 to 2022 as sample, we have discussed the impact of positive tone of annual report text on corporate green innovation and mechanism. We found that the positive tone of the annual report text plays a positive information incremental effect, significantly improves green innovation, and encourages enterprises to improve the quantity and quality of green innovation. Through the mechanism test, we found that the positive tone of the annual report text can improve green innovation mainly through the two paths of easing the financing constraints of enterprises and enhancing market attention. Heterogeneity analysis found that the positive tone of the annual report text had a more significant effect on the green innovation of high economic policy uncertainty and non-heavy polluting enterprises. Finally, we tested the economic effects on corporate value. We found that the positive tone of the annual report text on improving corporate green innovation is conducive to enhancing corporate value.

According to the research contents and conclusions, there are mainly three enlightenments as follows: Firstly, the positive tone of the corporate annual report text releases the signal of the future development of the enterprise and promotes the implementation of the green innovation strategy of the enterprise in both quantitative and qualitative levels. Therefore, enterprises should actively establish and improve the information transmission mechanism of information disclosure tone and provide capital market participants with text information conveying enterprises' high-quality, accurate, and objective development status. Capital market participants can reasonably use corporate annual report text information intonation to predict corporate green innovation strategy implementation. Secondly, the annual report is essential for investors to make value judgments and investment decisions. In the face of the complicated annual report, how to extract practical information and see the true face of the company is a problem that troubles many investors. The positive tone of the annual report text has information value and certain credibility, which can help investors quickly understand the company's performance and strategy implementation. Therefore, the regulatory authorities should increase the standardization of the disclosure system of corporate annual report text information and improve the true, accurate, and objective expression of text information better to serve the users of annual report information for decision-making. Thirdly, sufficient funds and high external attention are the driving forces for enterprises to implement green innovation. Therefore, the government should implement the regulations and practices of a unified large market and fair competition as soon as possible so that more enterprises can increase investment in green technology research and development in an honest, orderly, open, and transparent market competition environment, promote green transformation, and build a green development model.

This study has several limitations, including (i)the positive tone of the text is limited to management discussion and analysis in the annual report, (ii) other measures of green innovation are not used, (iii) a simple regression model is used, and (iv) the influence of external governance mechanisms on the study is not considered. Therefore, future research may use more channels of annual report text tone, use other green innovation measures, apply other suitable models, and consider the impact of multiple background factors on the research (e.g., media coverage, institutional investors, and independent directors).

## Supporting information

**S1 Data. Empirical data.**
(ZIP)

## Author Contributions

**Data curation:** Yange Gao.

**Formal analysis:** Yange Gao.

**Methodology:** Yange Gao.

**Project administration:** Jian Feng.

**Resources:** Yange Gao.

**Writing – original draft:** Yange Gao.

**Writing – review & editing:** Yange Gao, Jian Feng.

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
