## [Decision Letter · Decision Letter 0]

10 May 2024

PONE-D-24-12298Annual report text's positive tone and corporate green innovation: Evidence from ChinaPLOS ONE

Dear Dr. Gao,

Thank you for submitting your manuscript to PLOS ONE. After careful consideration, we feel that it has merit but does not fully meet PLOS ONE’s publication criteria as it currently stands. Therefore, we invite you to submit a revised version of the manuscript that addresses the points raised during the review process.

We look forward to receiving your revised manuscript.

Kind regards,

Jin Hu

Academic Editor

PLOS ONE

Journal Requirements:

   "N/A" 

3. For studies involving third-party data, we encourage authors to share any data specific to their analyses that they can legally distribute. PLOS recognizes, however, that authors may be using third-party data they do not have the rights to share. When third-party data cannot be publicly shared, authors must provide all information necessary for interested researchers to apply to gain access to the data. (https://journals.plos.org/plosone/s/data-availability#loc-acceptable-data-access-restrictions) 

Additional Editor Comments:

Thank you for your submission. The manuscript presents an interesting and relevant study on the impact of the positive tone in annual report text on corporate green innovation. The reviewers have provided constructive feedback, indicating several areas that require substantial revisions to improve the quality and clarity of the paper. Please carefully consider each reviewer's comments and address the concerns raised. If you choose to challenge or disagree with a reviewer's comment, ensure that your response is clear, logical, and supported by evidence.

When preparing your response to reviewers, please take the following into account:

1.Address each point raised by the reviewers, providing a detailed explanation of how you have resolved the issue or your rationale for disagreement.

2.Avoid including confidential information, items under copyright, or libelous or discriminatory statements in your response.

3.If you made significant changes to the manuscript, highlight those sections and explain how they enhance the paper.

The reviewers' key points for improvement include addressing endogeneity, enhancing theoretical discussions, updating the data to a more recent timeframe, improving the language and writing structure, correcting errors in equations, and adding more robust statistical analyses. Please find below the full list of comments from the reviewers and a summary of their concerns.

I look forward to receiving your revised manuscript along with your detailed response to the reviewers' comments. If you need additional time to complete the revisions, please let me know, and I will do my best to accommodate your request.

Thank you again for your submission, and I look forward to seeing your revisions.

Best regards,

Jin Hu

hujin@mail.gufe.edu.cn

Reviewers' comments:

Reviewer's Responses to Questions

**Comments to the Author**

1. Is the manuscript technically sound, and do the data support the conclusions?

Reviewer #1: Yes

Reviewer #2: Yes

Reviewer #3: Yes

2. Has the statistical analysis been performed appropriately and rigorously? 

Reviewer #1: Yes

Reviewer #2: Yes

Reviewer #3: No

3. Have the authors made all data underlying the findings in their manuscript fully available?

Reviewer #1: No

Reviewer #2: Yes

Reviewer #3: Yes

4. Is the manuscript presented in an intelligible fashion and written in standard English?

Reviewer #1: Yes

Reviewer #2: Yes

Reviewer #3: Yes

5. Review Comments to the Author

Reviewer #1: The study examines the impact of the positive tone in annual report text on corporate green innovation. Using a sample of listed companies from 2010 to 2020, the authors find a significant association between a positive tone in annual reports and enhanced green innovation, both in terms of quantity and quality. Mechanism tests indicate that this effect is primarily driven by the alleviation of corporate financing constraints and increased external attention. Heterogeneity analysis suggests a stronger effect in non-state-owned and non-heavily polluting industries. Additionally, the study concludes that a positive tone in annual reports ultimately contributes to the long-term value of companies through green innovation. Overall, the study contributes significantly to the theoretical understanding of annual report text tone and provides empirical evidence supporting the role of positive tone in promoting enterprise green innovation.

1. Equation (2): There are spelling errors in Equation (2), such as "cashflow" and "growth," and there are missing subscripts, such as "it."

2. Economic Effect Analysis: Please consider incorporating an economic effect analysis similar to the approach followed by Wang et al. (2024), taking into account multiple enterprise performance indicators.

3. Endogeneity Issue: I recommend employing the Generalized Method of Moments (GMM) to address the endogeneity issue present in the study.

4. Theoretical Hypotheses: Please include a discussion in the theoretical hypotheses section regarding how the positive tone in annual reports can positively influence enterprises' green innovation through the mechanisms of alleviating financing constraints and enhancing external attention, as suggested by previous literature (e.g., Wang et al., 2024).

Reference:

Wang et al. (2024) "Can green funds improve corporate environmental, social, and governance performance? Evidence from Chinese-listed companies" Plos One

Reviewer #2: The manuscript "PONE-D-24-12298" examines the relationship between the positive tone of annual report text and corporate green innovation. The content of the paper is rich and displays certain innovative aspects. To enhance the quality of the paper, the authors might consider addressing the following issues:

1. The introduction mentions, "Corporate managers may deliberately use a positive tone to mislead investors to cover up the poor performance of enterprises in environmental and social domains." Is this the motivation for the research? Has the manuscript addressed this issue? If so, how?

2. The language of the manuscript requires improvement. For example, sections such as "Section 2 literature review, theoretical analysis and hypothesis" and "Section 5 mechanism test and heterogeneity analysis" lack necessary sentence components.

3. The research hypotheses presented should be consistent with the conclusions that the paper aims to verify; therefore, one hypothesis should be removed.

4. The author seems to have a non-rigorous attitude towards academic writing, as seen in Section 3, "Following prior studies (),".

5. Why does the data only extend to 2020? What is the reason for this?

6. The practice of adding 1 to the data before taking logarithms, which has been considered as distorting the data and regression results, should be reevaluated. Why not consider using raw data for regression?

7. Have macroeconomic variables not been considered among the control variables?

8. There are unrecognizable symbols in Equation (2); please verify.

9. The descriptive statistics show that more than half of the data on green technology innovation are zeros. Is this a true reflection of the situation?

10. Is there a basis for the calculation of Tone2?

11. The section on research hypotheses could benefit from additional theoretical discussion related to the mechanism section.

12. The author needs to discuss the rationale and significance behind analyzing heterogeneity.

13. Results from the inter-group variance tests should be included in the heterogeneity analysis.

Reviewer #3: Q1 The number of green patent applications is a typical ordered count data, which does not obey the positive distribution. I do not recommend that authors use +1 and logarithmic data processing methods. I prefer to use Poisson regression or panel negative binomial regression.

Q2 I strongly recommend that the sample data be updated to 2022. In the past two years, China's capital market has continued to improve, while the digital transformation of enterprises has become more mature. These reasons may have implications for green innovation.

Q3 The study did not consider causal inference. Note that PSM does not solve the endogeneity problem. Regression using a one-stage lag of the explained variable is also not a causal inference method. I recommend that authors adopt the tool variable and Heckman two-step method.

Q4 Baseline regression. I strongly recommend that the authors adequately compare the results of this study with existing literature. For the researchers, we are eager to know whether this manuscript is the first to discover the positive effect of positive tone in annual reports on green innovation, or whether it is a re-examination of the existing literature.

Q5 Study limitations and future prospects are suggested to be supplemented.

Q6 I suggest that the author add a research flow chart to the introduction.

6. PLOS authors have the option to publish the peer review history of their article (what does this mean?). If published, this will include your full peer review and any attached files.

Reviewer #1: No

Reviewer #2: No

Reviewer #3: No

---

## [Author Response · Author response to Decision Letter 0]

16 May 2024

Dear Editors and Reviewers:

Thank you very much for your comments and professional advice concerning our manuscript entitled ''Annual report text's positive tone and corporate green innovation: Evidence from China'' (ID: PONE-D-24-12298). Those comments are all valuable and helpful for revising and improving our paper. We have studied the comments carefully and have revised our manuscript. Words in blue color are the changes we have made in the revised manuscript. In addition, since the sample time has been updated to 2022, the data in the table in the revised manuscript has also been updated accordingly. Now we respond to your comments point by point and highlight the changes in the revised manuscript. We sincerely hope that you find our responses and modifications satisfactory. The responses to your comments are listed as follows:

Reviewer # 1:

1.Comment: Equation (2): There are spelling errors in Equation (2), such as "cashflow" and "growth," and there are missing subscripts, such as "it." 

1. Response: Thank you for your comments and suggestions, we have modified the spelling errors in Equation 2, the revised Equation 2 as follows:

〖Green〗_(i,t)=α_0+α_1 〖Tone〗_(i,t)+α_2 S〖ize〗_(i,t)+α_3 〖Lev〗_(i,t)+α_4 〖ROA〗_(i,t)+α_5 〖Cashflow〗_(i,t)+α_6 〖Growth〗_(i,t)+α_7 〖Board〗_(i,t)+α_8 〖Indep〗_(i,t)+α_9 〖Top1〗_(i,t)+α_10 〖ListAge〗_(i,t)+α_11 〖Dual〗_(i,t)+α_12 〖SOE〗_(i,t)+α_13 〖Year〗_(i,t)+〖 α〗_14 〖Ind〗_(i,t)+ε_(i,t) (2) (Line 209-211 on page8)

2. Comment: Economic Effect Analysis: Please consider incorporating an economic effect analysis similar to the approach followed by Wang et al. (2024), taking into account multiple enterprise performance indicators. 

2. Response: Thank you for your suggestion. according to your suggestion, we have adopted an economic effects analysis method similar to that employed by Wang et al. (2024), and have considered multiple firm performance indicators simultaneously. The revised section as follows: 

''Since enterprises have invested a lot in the early stage of implementing a green innovation strategy, its economic benefits may not be reflected in the short term. Therefore, for corporate value indicators, following Wang et al. (2024), we adopt Tobin’s Q ratio (TobinQ) and the Price-to-Book ratio (PB) to measure, focusing on the impact on the long-term value of enterprises, and constructs the following Equation (4) to test the effect of positive tone of annual report text and green innovation on corporate value. The other control variables in Equation (4) are consistent with Equation (2), focusing on the coefficient of cross-term Tone*Green.

The regression results are shown in Table 12. In column (1) and (2), the coefficient of Tone*Green is significantly positive at a 1% level, indicating that the positive tone of the annual report text has a positive impact on corporate green innovation and can improve corporate value. ''(Line 418-420; 424 on page21) 

3. Comment: Endogeneity Issue: I recommend employing the Generalized Method of Moments (GMM) to address the endogeneity issue present in the study. 

3 Response: Your comments have been very helpful to us, according to your suggestion, we have adopted the instrumental variable method to solve the endogeneity problem, the revised section as follows:

''4.5.6 Instrumental variable method

Considering that there may be a reverse causal relationship between the positive tone of annual report texts and green innovation, following Li and Jiang (2020), we select the mean value of the positive tone of texts in the same year and industry as the instrumental variable (Tone_m) and adopt the two-stage least squares method (IV-2SLS). 

The regression results are shown in Table 8. The test results of weak instrumental variables show that the F value is 3054.94, which is significantly greater than the empirical value 10. The instrumental variables satisfy the correlation and are strong instrumental variables. The regression coefficient of Tone is still significantly positive, which once again verifies the positive tone of the annual report text to promote enterprises' green innovation.'' (Line 309-319 on page15)

Table 8 Instrumental variable method

Variable Green

 (1) (2)

Tone_m 1.3852** 

 (2.0597) 

Tone 1.7996**

 (2.0609)

Size 0.1372*** 0.1364***

 (33.8924) (33.2899)

Lev 0.1407*** 0.1352***

 (5.6741) (5.4336)

ROA 0.4021*** 0.3346***

 (5.5959) (4.2046)

Cashflow 0.0840 0.0920

 (1.3763) (1.5049)

Growth -0.0509*** -0.0554***

 (-5.3560) (-5.6942)

Board 0.0765*** 0.0742***

 (3.1126) (3.0151)

Indep 0.0009 0.0009

 (1.1232) (1.1035)

Top1 -0.0005* -0.0005*

 (-1.7100) (-1.6592)

ListAge -0.0812*** -0.0792***

 (-13.9104) (-13.2109)

Dual 0.0177** 0.0171*

 (2.0134) (1.9424)

SOE 0.0814*** 0.0763***

 (8.1450) (7.2863)

_cons -2.9496*** -2.9495***

 (-9.2057) (-9.2108)

Year fe Yes Yes

Ind fe Yes Yes

F-value 3054.94

N 34799 34799

r2 0.1754 0.1764

Note: ***p<0.01, **p<0.05, *p<0.1 are significant at the level of 1%, 5% and 10%, respectively.''

4. Comment: Theoretical Hypotheses: Please include a discussion in the theoretical hypotheses section regarding how the positive tone in annual reports can positively influence enterprises' green innovation through the mechanisms of alleviating financing constraints and enhancing external attention, as suggested by previous literature (e.g., Wang et al., 2024). 

4. Response: Thank you for your suggestion, according to your suggestion, we have added relevant literature research to the revised manuscript. We hope that the current theoretical analysis is more reasonable, and the revised section is as follows:

'' This enables investors to have a deeper understanding of the economic conditions of enterprises and increase their investment in enterprises (Feldman et al., 2010; Liu and Wang, 2021), to reduce financing constraints (Qiu and Yang, 2021; Wang et al., 2024), effectively solve the financing dilemma of enterprises' green innovation, and improve the willingness of enterprises to green innovation. '' (Line 131 on page 5)

'' In the current wave of green and low-carbon development, the public's environmental demands for green and low-carbon, ecological, and environmental protection are important driving forces for enterprises to carry out green innovation and development and achieve green transformation (Wang et al., 2024). Therefore, under the supervision and pressure of external analysts, enterprises will actively implement green innovation activities to meet the demands of the public. (Chen et al., 2018; Wang et al., 2024). '' (Line 139-141 on page 5)

The above contents are our responses to your valuable questions or suggestions. Thank you very much for your patience! Your patience and high-quality advice are helpful to improve our paper.

Reviewer # 2:

1. Comment: The introduction mentions, "Corporate managers may deliberately use a positive tone to mislead investors to cover up the poor performance of enterprises in environmental and social domains." Is this the motivation for the research? Has the manuscript addressed this issue? If so, how? 

1. Response: Thank you for pointing out this issue. as for the role of the positive tone of the annual report text, the academic community has drawn two opposite views. There are positive and negative views on the role of positive tone in the annual report text. Then, when it comes to the impact on green innovation, does positive text intonation play a positive information incremental effect or a negative information masking effect? This is also the motivation of this study. After empirical testing of samples from 2010 to 2022, this study concluded that positive intonation can promote enterprises to carry out green innovation behaviors, supporting the information incremental view. And further research shows that positive tone can improve the level of green innovation and enterprise value at the same time, which further proves the positive role of positive tone.

2. Comment: The language of the manuscript requires improvement. For example, sections such as "Section 2 literature review, theoretical analysis and hypothesis" and "Section 5 mechanism test and heterogeneity analysis" lack necessary sentence components. 

2. Response: We feel so sorry for the mistakes in the manuscript and inconvenience they caused in your reading. The language in our revised manuscript has been professionally edited and verified by a native English speaker. Thank you very much for your precious comments. In addition, we have improved our presentation by clearly labeling equations, tables, sections, and subsections. We really hope that the flow and language level have been substantially improved. 

3. Comment: The research hypotheses presented should be consistent with the conclusions that the paper aims to verify; therefore, one hypothesis should be removed.

3. Response: Thank you very much for your suggestions. Please allow us for some explanation about this problem. There are positive and negative views on the role of positive tone in the annual report text. The research perspective of this paper is green innovation, from the theoretical level, we can deduce two opposite logical paths. Before the empirical test, we do not know whether it is positive or negative, and only through the subsequent empirical test can we reach the final conclusion. Therefore, we put forward two competing hypotheses in the theoretical analysis and hypothesis part.

4. Comment: The author seems to have a non-rigorous attitude towards academic writing, as seen in Section 3, "Following prior studies (),". 

4. Response: Thank you for pointing out this issue. We feel so sorry for the mistakes in the manuscript. We have revised this question, as well as other related questions throughout the manuscript. The modified section is as follows:

'' Following prior studies (Xu et al., 2020; Yu, 2022)'' (Line 543-551 on page 24).'' 

5. Comment: Why does the data only extend to 2020? What is the reason for this?

5. Response: Thank you for pointing out this issue. We have updated the sample data to 2022.

6. Comment: The practice of adding 1 to the data before taking logarithms, which has been considered as distorting the data and regression results, should be reevaluated. Why not consider using raw data for regression? 

6. Response: Thank you for pointing out this question. After descriptive statistics, we found that the green innovation data was obviously skewed to the right. Therefore, we followed the practice of previous studies and processed the green innovation data with logarithm after +1. At the same time, in the robust test part, we also used the original data of green innovation to conduct a re-regression test, and the research conclusion remained unchanged.

7. Comment: Have macroeconomic variables not been considered among the control variables? 

7. Response: We did not take macro variables as control variables for two reasons: First, because the problem we studied was at the micro enterprise level, it was more affected by corporate governance and financial characteristics. Second, the selection of control variables in this study also follows the practice of previous studies.

8. Comment: There are unrecognizable symbols in Equation (2); please verify. 

8. Response: Thank you for pointing out this question. The revised Equation (2) is as follows: 

〖Green〗_(i,t)=α_0+α_1 〖Tone〗_(i,t)+α_2 S〖ize〗_(i,t)+α_3 〖Lev〗_(i,t)+α_4 〖ROA〗_(i,t)+α_5 〖Cashflow〗_(i,t)+α_6 〖Growth〗_(i,t)+α_7 〖Board〗_(i,t)+α_8 〖Indep〗_(i,t)+α_9 〖Top1〗_(i,t)+α_10 〖ListAge〗_(i,t)+α_11 〖Dual〗_(i,t)+α_12 〖SOE〗_(i,t)+α_13 〖Year〗_(i,t)+〖 α〗_14 〖Ind〗_(i,t)+ε_(i,t) (2) (Line 209-211 on page8)

9. Comment: The descriptive statistics show that more than half of the data on green technology innovation are zeros. Is this a true reflection of the situation? 

9. Response: Thank you for pointing out this issue. This is a true reflection of the situation, and previous studies have come to the same result.

10. Comment: Is there a basis for the calculation of Tone2? 

10. Response: Thank you very much for pointing out our problem. For the measurement of Tone2, the intonation library built is the same as Tone, but different calculation formulas are adopted. The specific formula is shown in Equation (3). Therefore, in the original manuscript, we simplified the process and did not describe the construction of the intonation library again. We have re-described the Tone2 measurement process in detail in the revised manuscript, hoping that our revisions can make the manuscript more clear. The revised contents are as follows.

''To overcome the influence of the measurement bias of explanatory variables, Following Xie and Lin (2015), first of all, we take positive color words (" positive, "high quality," "promotion," etc.) and negative color words (" low, "unfavorable," "downward," etc.) as a list of positive and negative words respectively. Then, we used Python programming to count the positive and negative words in the MD&A of the annual report to obtain the total number of positive and negative words. Finally, refering to the practice of Li and Jiang (2020), we use the Equation (3) to calculate the report text's positive tone (Tone2).'' (Line 265-269 on page12)

11. Comment: The section on research hypotheses could benefit from additional theoretical discussion related to the mechanism section. 

11. Response: In the part of theoretical analysis and hypothesis, we put forward two opposing hypotheses based on the two logic chains of information increment view and impression management view. But which hypothesis is valid? After the empirical test, our conclusion validates the information increment view, that is, the positive tone of the annual report text can promote the green innovation behavior of enterprises.Therefore, in the latter part of the mechanism test, we need to find out the channel through which the positive tone of the annual report text promotes the green innovation of enterprises under the theoretical framework of the incremental view of information (theoretical analysis and hypothesis).

12. Comment: The author needs to discuss the rationale and significance behind analyzing heterogeneity. 

12. Response: Thank you for your suggestion, according to your suggestion, we rewrote the heterogeneity analysis section. The rewritten content is as follows:

''5.2.1 The influence of economic policy uncertainty

China, transitioning from high-speed economic growth to high-quality development, faces significantly more risks and challenges, and its economic development has significant destabilizing and uncertain factors. To cope with the downward economic pressure brought about by risks and challenges, China has continuously carried out macroeconomic policy reform and innovation, and the uncertainty of economic policy has increased significantly (Chen and Chen, 2018), which not only directly affects the efficiency and effect of the macroeconomic operation, but also spreads to micro-enterprises through the capital market mechanism. Finally, it will become an important external force affecting the green innovation of micro-enterprises. The rise of economic policy uncertainty will exacerbate market information asymmetry and increase enterprises' financing difficulty, thus inhibiting enterprises' green innovation (Drobetz et al., 2018; Arellano et al., 2019; Song et al., 2019). The positive tone of the annual report can reduce the information asymmetry between enterprises and external markets, thus easing the financing constraints of enterprises and providing funds for enterprises to carry out green innovation. Therefore, we predict that the positive tone of the annual report text will promote corporate green innovation more significantly in listed companies with high economic policy uncertainty.

 Following Ding and Wang (2021), we adopted the uncertainty index of China's economic policy constructed by Baker et al. (2016) based on the South China Morning Post, a news report retrieval platform, through keyword search. Then, the logarithm of t

---

## [Decision Letter · Decision Letter 1]

24 May 2024

PONE-D-24-12298R1Annual report text's positive tone and corporate green innovation: Evidence from ChinaPLOS ONE

Dear Dr. Gao,

Thank you for submitting your manuscript to PLOS ONE. After careful consideration, we feel that it has merit but does not fully meet PLOS ONE’s publication criteria as it currently stands. Therefore, we invite you to submit a revised version of the manuscript that addresses the points raised during the review process.

We look forward to receiving your revised manuscript.

Kind regards,

Jin Hu

Academic Editor

PLOS ONE

Journal Requirements:

Reviewers' comments:

Reviewer's Responses to Questions

**Comments to the Author**

1. If the authors have adequately addressed your comments raised in a previous round of review and you feel that this manuscript is now acceptable for publication, you may indicate that here to bypass the “Comments to the Author” section, enter your conflict of interest statement in the “Confidential to Editor” section, and submit your "Accept" recommendation.

Reviewer #1: All comments have been addressed

Reviewer #2: (No Response)

Reviewer #3: All comments have been addressed

2. Is the manuscript technically sound, and do the data support the conclusions?

Reviewer #1: Yes

Reviewer #2: Partly

Reviewer #3: Yes

3. Has the statistical analysis been performed appropriately and rigorously? 

Reviewer #1: Yes

Reviewer #2: Yes

Reviewer #3: Yes

4. Have the authors made all data underlying the findings in their manuscript fully available?

Reviewer #1: Yes

Reviewer #2: (No Response)

Reviewer #3: Yes

5. Is the manuscript presented in an intelligible fashion and written in standard English?

Reviewer #1: Yes

Reviewer #2: No

Reviewer #3: Yes

6. Review Comments to the Author

**Reviewer #1: **Please pay attention to the formatting of the formula. For instance, in formula (2) should not be on a separate line, and formula (3) should be centered.

**Reviewer #2:** Thank you for the efforts made in revising the manuscript PONE-D-24-12298. However, I am dissatisfied with the outcome of the previous version's revisions as the feedback provided was not adequately addressed. The reasons for my dissatisfaction are as follows:

1. The inter-group variance tests for heterogeneity analysis are missing. Where have they been conducted?

2. The presence of micro-level data does not justify overlooking macro-level influences. Many micro-studies control for macroeconomic variables.

3. Equations (1), (2), and (3) are not displayed on the same line. The author needs to approach the manuscript revision with greater seriousness.

4. The author has not amended the following incomplete sentence which lacks a predicate: "Section 2 literature review, theoretical analysis, and hypothesis. Section 5 mechanism test and heterogeneity analysis." Is the author taking the review process lightly?？

5. I suggest the author thoroughly proofread the manuscript before resubmitting！

**Reviewer #3: **The author did a good job of solving my earlier doubts. On the whole, I am satisfied with the revised manuscript. Before accepting this manuscript, however, I recommend that the authors intensify the literature review. I highly recommend adding references from the last three years to highlight the timeliness of the manuscript. I have listed some of the literature, and authors can decide whether they need to cite them based on the relevance of the research.

Hu, J., Hu, M., & Zhang, H. (2023). Has the construction of ecological civilization promoted green technology innovation?. Environmental Technology & Innovation, 29, 102960.

Bai, D.B., Hu, J., Irfan, M., & Hu, M.J. (2023). Unleashing the impact of ecological civilization pilot policies on green technology innovation: Evidence from a novel SC-DID model. Energy Economics, 106813.

Han, Y., Li, Z., Feng, T., Qiu, S., Hu, J., Yadav, K. K., & Obaidullah, A. J. (2024). Unraveling the impact of digital transformation on green innovation through microdata and machine learning. Journal of Environmental Management, 354, 120271.

Shen, Y., Zhang, X. (2023). Intelligent manufacturing, green technological innovation and environmental pollution. Journal of Innovation & Knowledge 8(3), 100384.

Finally, I suggest replacing the table in Table 3 with a picture. Thermodynamic coefficient is recommended.

7. PLOS authors have the option to publish the peer review history of their article (what does this mean?). If published, this will include your full peer review and any attached files.

Reviewer #1: No

Reviewer #2: No

Reviewer #3: No

---

## [Author Response · Author response to Decision Letter 1]

27 May 2024

Dear Editors and Reviewers:

Thank you very much for your comments and professional advice concerning our manuscript entitled ''Annual report text's positive tone and corporate green innovation: Evidence from China'' (ID: PONE-D-24-12298). Those comments are all valuable and helpful for revising and improving our paper. We have studied the comments carefully and have revised our manuscript. Words in blue color are the changes we have made in the revised manuscript. Now we respond to your comments point by point and highlight the changes in the revised manuscript. We sincerely hope that you find our responses and modifications satisfactory. The responses to your comments are listed as follows:

Reviewer # 1:

1.Comment: Please pay attention to the formatting of the formula. For instance, in formula (2) should not be on a separate line, and formula (3) should be centered.

1. Response: Thank you for your comments and suggestions, we have modified formula (2) to ensure that the format is standardized. the revised Equation 2 as follows:

〖 Green〗_(i,t)=α_0+α_1 〖Tone〗_(i,t)+α_j ∑▒〖Controls〗_(i,t) +∑▒〖Year〗_(i,t) +∑▒〖Ind〗_(i,t) +ε_(i,t) (2)

(Line 209 on page8)

The above contents are our responses to your valuable questions or suggestions. Thank you very much for your patience! Your patience and high-quality advice are helpful to improve our paper.

Reviewer # 2:

1. Comment: The inter-group variance tests for heterogeneity analysis are missing. Where have they been conducted? 

1. Response: Thank you very much for asking this question, but we have to explain this, for the coefficient difference between groups, we've done this in the original manuscript using the Bootstrap sampling method to test the significance of inter-group coefficient differences. Moreover, it is also explained in the paper, including the method and p-value, and the P-value is also shown in the table. The specific contents of the inter-group coefficient difference test are as follows:

'' However, when we further used the Bootstrap sampling method to test the significance of inter-group coefficient differences, the corresponding P-value was 0.000, indicating that when the economic policy uncertainty was higher, the positive tone of the annual report had a more obvious role in promoting corporate green innovation. This test result further confirmed the information incremental effect of positive tone of the text. '' (Line 384-486 on page19)

Table 11 The influence of economic policy uncertainty

Variable Green

 (1)

EPU_H (2)

EPU_L

Tone 2.5714*** 0.9124***

 (6.2380) (3.0190)

Size 0.1471*** 0.1316***

 (17.1074) (19.4592)

Lev 0.1554*** 0.1105***

 (4.1084) (3.8429)

ROA 0.2113** 0.4864***

 (2.1558) (5.2977)

Cashflow 0.1457 0.0511

 (1.3894) (0.7594)

Growth -0.0578*** -0.0565***

 (-3.1368) (-7.0944)

Board 0.0502 0.0921***

 (1.1079) (2.6060)

Indep 0.0007 0.0010

 (0.4558) (0.9090)

Top1 -0.0002 -0.0006*

 (-0.3215) (-1.7762)

ListAge -0.0947*** -0.0699***

 (-10.0420) (-9.4920)

Dual 0.0119 0.0211*

 (0.8427) (1.8371)

SOE 0.0858*** 0.0719***

 (4.8841) (5.8116)

_cons -3.2101*** -3.1034***

 (-12.5828) (-19.4685)

Year fe Yes Yes

Ind fe Yes Yes

p-value 0.000

N 14539 20291

r2 0.1820 0.1737

'' When we further used the Bootstrap sampling method to test the significance of coefficient differences between groups again, the corresponding P value was 0.030, indicating that the positive impact of the positive tone of annual report text on enterprise green innovation was more significant in non-heavy pollution enterprises. ''(Line 409-410 on page21)

Table 12 The influence of industry attributes

Variable Green

 (1)

Heavy-polluting industries (2)

Non-heavy polluting industries

Tone 0.9926** 1.8604***

 (2.2232) (6.3929)

Size 0.1251*** 0.1388***

 (15.3121) (21.4571)

Lev -0.1300*** 0.2124***

 (-2.8428) (8.0826)

ROA 0.0166 0.4077***

 (0.1184) (5.6243)

Cashflow 0.3876*** 0.0133

 (3.4582) (0.1992)

Growth -0.0371** -0.0586***

 (-2.3156) (-6.6983)

Board 0.2627*** 0.0264

 (4.9257) (0.8153)

Indep 0.0048** -0.0002

 (2.5740) (-0.2030)

Top1 0.0007 -0.0007**

 (1.2346) (-2.0890)

ListAge -0.0736*** -0.0818***

 (-6.3576) (-12.1831)

Dual -0.0440*** 0.0320***

 (-2.7101) (3.0687)

SOE 0.1037*** 0.0671***

 (5.4438) (5.6458)

_cons -3.0160*** -3.0908***

 (-10.8181) (-20.1513)

Year fe Yes Yes

Ind fe Yes Yes

p-value 0.030

N 7892 26938

r2 0.1411 0.1861

2. Comment: The presence of micro-level data does not justify overlooking macro-level influences. Many micro-studies control for macroeconomic variables. 

2. Response: Thank you very much for your questions and suggestions. According to your suggestions, after adding macroeconomic variables, we re-regression model (2), the research results remain unchanged, and the new contents are as follows:

'' 4.5.7 Adding macro variables

To make the results more robust, we added a macroeconomic variable, the value-added rate of gross domestic product (GDP), as a control variable. The regression results after adding macroeconomic variables are shown in Table9. The regression coefficient of Tone is 1.6409, which is significant at the 1% level, indicating that our main regression results are still robust. '''(Line 320-325 on page16)

Table 9 Robust test after adding macro variables

Variable Green

 (1)

Tone 1.6409***

 (6.6681)

Size 0.1360***

 (25.5848)

Lev 0.1385***

 (6.0557)

ROA 0.3509***

 (5.4223)

Cashflow 0.0925

 (1.5993)

Growth -0.0562***

 (-7.3296)

Board 0.0703**

 (2.5238)

Indep 0.0008

 (0.9061)

Top1 -0.0005*

 (-1.7102)

ListAge -0.0813***

 (-13.7965)

Dual 0.0166*

 (1.8508)

SOE 0.0789***

 (7.7456)

GDP -0.0160***

 (-6.4435)

_cons -2.7560***

 (-21.4502)

Year fe Yes

Ind fe Yes

N 34664

r2 0.1769

3. Comment: Equations (1), (2), and (3) are not displayed on the same line. The author needs to approach the manuscript revision with greater seriousness. ✅

3. Response: Thank you for your comments and suggestions, to ensure that equations (1), (2) and (3) have a consistent format, we have modified Equation 2. the revised Equation 2 as follows:

〖 Green〗_(i,t)=α_0+α_1 〖Tone〗_(i,t)+α_j ∑▒〖Controls〗_(i,t) +∑▒〖Year〗_(i,t) +∑▒〖Ind〗_(i,t) +ε_(i,t) (2)

(Line 209 on page8)

4. Comment: The author has not amended the following incomplete sentence which lacks a predicate: "Section 2 literature review, theoretical analysis, and hypothesis. Section 5 mechanism test and heterogeneity analysis." Is the author taking the review process lightly?

4. Response: We feel so sorry for the mistakes in the manuscript and inconvenience they caused in your reading. The language in our revised manuscript has been professionally edited and verified by a native English speaker. Thank you very much for your precious comments. We take every question you raise very seriously and try our best to revise the manuscript. Thank you very much for the time and effort you have put into the manuscript. We really hope that the flow and language level have been substantially improved. 

5. Comment: I suggest the author thoroughly proofread the manuscript before resubmitting！

5. Response: Thank you very much for your suggestion. We have carried out a comprehensive examination of the manuscript and hope to get your approval.

The above contents are our responses to your valuable questions or suggestions. Your high-quality suggestions are of great help in improving the quality of our paper. Special thanks to you for your good comments!

Reviewer # 3:

1. Comment: The author did a good job of solving my earlier doubts. On the whole, I am satisfied with the revised manuscript. Before accepting this manuscript, however, I recommend that the authors intensify the literature review. I highly recommend adding references from the last three years to highlight the timeliness of the manuscript. I have listed some of the literature, and authors can decide whether they need to cite them based on the relevance of the research.

Hu, J., Hu, M., & Zhang, H. (2023). Has the construction of ecological civilization promoted green technology innovation?. Environmental Technology & Innovation, 29, 102960.

Bai, D.B., Hu, J., Irfan, M., & Hu, M.J. (2023). Unleashing the impact of ecological civilization pilot policies on green technology innovation: Evidence from a novel SC-DID model. Energy Economics, 106813.

Han, Y., Li, Z., Feng, T., Qiu, S., Hu, J., Yadav, K. K., & Obaidullah, A. J. (2024). Unraveling the impact of digital transformation on green innovation through microdata and machine learning. Journal of Environmental Management, 354, 120271.

Shen, Y., Zhang, X. (2023). Intelligent manufacturing, green technological innovation and environmental pollution. Journal of Innovation & Knowledge 8(3), 100384.

1. Response: Thank you very much for your suggestion, we have added to the literature according to your suggestion. The changes are as follows:

'' Green innovation, as a combination of the two new development concepts of "green" and "innovation," has been widely recognized and supported by the public (Hu et al., 2023; Bai et al., 2023). However, compared with the general technological innovation, the natural high risk, uncertainty, and long-term cyclical characteristics of green innovation will have a superimposed impact on the business risk of enterprises, which may lead to the lack of internal drive for green innovation (Han et al., 2024; Shen and Zhang, 2023). '' (Line 108, 111-112 on page 4)

2. Comment: Finally, I suggest replacing the table in Table 3 with a picture. Thermodynamic coefficient is recommended. 

2. Response: Thank you for your suggestions. According to your suggestion, we have converted Table 3 into a picture, as follows: (Line232 on page9)

We appreciate for Editors and Reviewers’ warm work earnestly, and we tried our best to improve the manuscript and made some changes in the manuscript. We hope that the correction will meet with approval. 

Thanks again for your helpful comments and suggestions! Best wishes to you!

---

## [Decision Letter · Decision Letter 2]

4 Jun 2024

PONE-D-24-12298R2Annual report text's positive tone and corporate green innovation: Evidence from ChinaPLOS ONE

Dear Dr. Gao,

Thank you for submitting your manuscript to PLOS ONE. After careful consideration, we feel that it has merit but does not fully meet PLOS ONE’s publication criteria as it currently stands. Therefore, we invite you to submit a revised version of the manuscript that addresses the points raised during the review process.

We look forward to receiving your revised manuscript.

Kind regards,

Jin Hu

Academic Editor

PLOS ONE

Reviewers' comments:

Reviewer's Responses to Questions

**Comments to the Author**

1. If the authors have adequately addressed your comments raised in a previous round of review and you feel that this manuscript is now acceptable for publication, you may indicate that here to bypass the “Comments to the Author” section, enter your conflict of interest statement in the “Confidential to Editor” section, and submit your "Accept" recommendation.

Reviewer #1: All comments have been addressed

Reviewer #2: (No Response)

Reviewer #3: All comments have been addressed

2. Is the manuscript technically sound, and do the data support the conclusions?

Reviewer #1: Yes

Reviewer #2: (No Response)

Reviewer #3: Yes

3. Has the statistical analysis been performed appropriately and rigorously? 

Reviewer #1: Yes

Reviewer #2: (No Response)

Reviewer #3: Yes

4. Have the authors made all data underlying the findings in their manuscript fully available?

Reviewer #1: No

Reviewer #2: (No Response)

Reviewer #3: Yes

5. Is the manuscript presented in an intelligible fashion and written in standard English?

Reviewer #1: Yes

Reviewer #2: (No Response)

Reviewer #3: Yes

6. Review Comments to the Author

Reviewer #1: (No Response)

Reviewer #2: The author still hasn't solved my problem. The sentence in lines 78-83 in the manuscript lacks the necessary verb, and I've already mentioned the same problem twice！

Reviewer #3: I am satisfied with the revised manuscript. Before accepting it, I suggest that the author revise the following details.

Q1 Correlation coefficient matrix can be used in heat maps. Display the picture in one visual and color, rather than simply replacing it with an image format.

Q2 Cite Chinese references as little as possible;

Q3 I suggest that the authors supplement the statement that instrumental variables satisfy the principles of relevance and exclusivity. Why does it have such a strong exogeneity?

Q4 Whether the result reported in parentheses is a t statistic or a standard error needs to be explained in a footnote.

7. PLOS authors have the option to publish the peer review history of their article (what does this mean?). If published, this will include your full peer review and any attached files.

Reviewer #1: No

Reviewer #2: No

Reviewer #3: No

---

## [Author Response · Author response to Decision Letter 2]

5 Jun 2024

Dear Editors and Reviewers:

Thank you very much for your comments and professional advice concerning our manuscript entitled ''Annual report text's positive tone and corporate green innovation: Evidence from China'' (ID: PONE-D-24-12298). Those comments are all valuable and helpful for revising and improving our paper. We have studied the comments carefully and have revised our manuscript. Words in blue color are the changes we have made in the revised manuscript. Now we respond to your comments point by point and highlight the changes in the revised manuscript. We sincerely hope that you find our responses and modifications satisfactory. The responses to your comments are listed as follows:

Reviewer # 2:

1. Comment: The author still hasn't solved my problem. The sentence in lines 78-83 in the manuscript lacks the necessary verb, and I've already mentioned the same problem twice！

1. Response: Thank you very much for your careful guidance in pointing out the problem of grammatical errors in the article. I am very sorry for the trouble caused to your reading. We have made careful revisions to address the language issues raised and hope to have your approval. The revisions are as follows:

'' The remainder of this study is organized as follows: Section 2 reviews the extant literature, discusses the theoretical framework of hypotheses and the potential channel in detail. Section 3 outlines the definitions of data and variables, as well as the construction of models. Section 4 presents the results and discussion, addresses the issue of endogeneity and robustness tests, and investigates the potential channels. Section 5 conducts mechanism tests, heterogeneity analysis and economic effects tests. Section 6 summarizes the conclusions and discusses the insights, limitations and future perspectives. This paper's research flow chart is shown in Figure 1.''(Line 78-84 on page3)

The above contents are our responses to your valuable questions or suggestions. Your high-quality suggestions are of great help in improving the quality of our paper. Special thanks to you for your good comments!

Reviewer # 3:

1. Comment: Correlation coefficient matrix can be used in heat maps. Display the picture in one visual and color, rather than simply replacing it with an image format.

1. Response: Thank you very much for your careful guidance. We have done the heat map as you suggested. We need to explain to you that we have not done a heat map before because the area of study has not seen such a map, which is usually presented in a table. So, we hope that the revised diagram meets your requirements. The revisions are as follows: (Line 232 on page 9)

Figure 2 Correlation analysis heat map

2. Comment: Cite Chinese references as little as possible

2. Response: Thank you for your suggestions. According to your suggestion, we have deleted some Chinese literature.

3. Comment: I suggest that the authors supplement the statement that instrumental variables satisfy the principles of relevance and exclusivity. Why does it have such a strong exogeneity?

3. Response: Thank you for your suggestions. According to your suggestion, we provide a detailed representation of the correlation and exogeneity of the instrumental variables. The revisions are as follows: 

''We adopt this idea of instrumental variables mainly for the following two reasons: Firstly, enterprises within the same industry have similar operation and business scope, and will be affected by the same market environment and regulatory rules. Therefore, the narrative logic and expression of the annual report texts of enterprises in the same industry tend to be more similar, presenting certain industry characteristics, which meets the relevance requirements of the instrumental variables. Secondly, the language characteristics of other enterprises' annual report texts do not have an impact on this enterprise's green innovation, which meets the requirement of exogeneity of instrumental variables. '' (Line315-322 on page15)

4. Comment: Whether the result reported in parentheses is a t statistic or a standard error needs to be explained in a footnote.

4. Response: Thank you for your suggestions. According to your suggestion, we have detailed notes underneath all the tables.

We appreciate for Editors and Reviewers’ warm work earnestly, and we tried our best to improve the manuscript and made some changes in the manuscript. We hope that the correction will meet with approval. 

Thanks again for your helpful comments and suggestions! Best wishes to you!

---

## [Decision Letter · Decision Letter 3]

13 Jun 2024

Annual report text's positive tone and corporate green innovation: Evidence from China

PONE-D-24-12298R3

Dear Dr. Gao,

We’re pleased to inform you that your manuscript has been judged scientifically suitable for publication and will be formally accepted for publication once it meets all outstanding technical requirements.

Kind regards,

Jin Hu

Academic Editor

PLOS ONE

Additional Editor Comments (optional):

Reviewers' comments:

Reviewer's Responses to Questions

**Comments to the Author**

1. If the authors have adequately addressed your comments raised in a previous round of review and you feel that this manuscript is now acceptable for publication, you may indicate that here to bypass the “Comments to the Author” section, enter your conflict of interest statement in the “Confidential to Editor” section, and submit your "Accept" recommendation.

Reviewer #1: (No Response)

Reviewer #2: (No Response)

Reviewer #3: All comments have been addressed

2. Is the manuscript technically sound, and do the data support the conclusions?

Reviewer #1: (No Response)

Reviewer #2: (No Response)

Reviewer #3: Yes

3. Has the statistical analysis been performed appropriately and rigorously? 

Reviewer #1: (No Response)

Reviewer #2: (No Response)

Reviewer #3: Yes

4. Have the authors made all data underlying the findings in their manuscript fully available?

Reviewer #1: (No Response)

Reviewer #2: (No Response)

Reviewer #3: Yes

5. Is the manuscript presented in an intelligible fashion and written in standard English?

Reviewer #1: (No Response)

Reviewer #2: (No Response)

Reviewer #3: Yes

6. Review Comments to the Author

Reviewer #1: (No Response)

Reviewer #2: (No Response)

Reviewer #3: The authors solved my doubts very well. I have no additional comments to offer, but suggest that the author carefully check the language and syntax of the full text to avoid false semantics. Now it's acceptable. congratulations

7. PLOS authors have the option to publish the peer review history of their article (what does this mean?). If published, this will include your full peer review and any attached files.

Reviewer #1: No

Reviewer #2: No

Reviewer #3: No

---

## [Editor Report · Acceptance letter]

21 Jun 2024

PONE-D-24-12298R3 

PLOS ONE

Dear Dr. Gao, 

I'm pleased to inform you that your manuscript has been deemed suitable for publication in PLOS ONE. Congratulations! Your manuscript is now being handed over to our production team.

Kind regards, 

on behalf of

Dr. Jin Hu 

Academic Editor

PLOS ONE